# An analytical theory of balanced cellular growth

Hugo Dourado [1] & Martin J. Lercher [1✉]

The biological fitness of microbes is largely determined by the rate with which they replicate their biomass composition. Mathematical models that maximize this balanced growth rate while accounting for mass conservation, reaction kinetics, and limits on dry mass per volume are inevitably non-linear. Here, we develop a general theory for such models, termed Growth Balance Analysis (GBA), which provides explicit expressions for protein concentrations, fluxes, and growth rates. These variables are functions of the concentrations of cellular components, for which we calculate marginal fitness costs and benefits that are related to metabolic control coefficients. At maximal growth rate, the net benefits of all concentrations are equal. Based solely on physicochemical constraints, GBA unveils fundamental quantitative principles of cellular resource allocation and growth; it accurately predicts the relationship between growth rates and ribosome concentrations in *E. coli* and yeast and between growth rate and dry mass density in *E. coli*.

[1] Institute for Computer Science & Department of Biology, Heinrich Heine University, 40221 Düsseldorf, Germany. ✉email: martin.lercher@hhu.de

The defining feature of life is self-replication. For non-interacting unicellular organisms in constant environments, the rate of this self-replication is equivalent to their evolutionary fitness[1]: fast-growing cells outcompete those growing more slowly. Accordingly, we expect that natural selection favoring fast growth in specific environments has played an important role in shaping the physiology of many microbial organisms[2,3].

Conceptually, we can envision a bacterial cell as a volume enclosed by a membrane, filled with a solution of metabolites and of the proteins and nucleic acids that catalyze their conversion into biomass. A state of the cell is characterized by the molecular concentrations, which in turn determine the fluxes of the biochemical reactions through kinetic rate laws. The boundary conditions limiting the concentrations and fluxes are provided by the environment and by physicochemical constraints. Cellular growth has to be balanced over the cell cycle, i.e., all cellular components must be produced in proportion to their abundances[4]. Casting these constraints into a mathematical model and characterizing states of optimal growth may provide a detailed understanding of central aspects of bacterial physiology[3,5–10].

Molenaar et al.[5] proposed a small, schematic model of balanced, self-replicating growth with explicit non-linear reaction kinetics and at most seven reactions, including the production of catalytic proteins. Numerical growth rate optimization predicted qualitatively the growth-rate dependencies of cellular ribosome content, cell size, and the emergence of overflow metabolism. We term this general modeling scheme Growth Balance Analysis (GBA). No extensions of this approach to larger models have been proposed, likely because of its inherent non-linearity and the resulting difficulty of numerical optimizations. Instead, even simpler, linear models of 1–3 reactions were solved analytically to gain further qualitative understanding of systems-level effects[3,6–9], including optimal gene regulation strategies[3,8].

Models for the genome-scale physiology of complete cells are typically formulated as approximations to GBA[11]. Currently, the most popular such method is flux balance analysis (FBA)[12,13]. FBA maximizes the production rate of a constant biomass concentration vector while accounting for mass conservation by balancing the fluxes producing and consuming internal metabolites (Fig. 1). All constraints in FBA are linear. The resulting computational efficiency comes at the price of ignoring reaction kinetics and the requirement of sufficient enzyme concentrations to catalyze the predicted metabolic fluxes. FBA can be viewed as a linearization of the GBA scheme[11]. Figure 1 shows a schematic comparison of FBA and GBA. While FBA predicts a linear dependence of maximal growth rate on nutrient uptake fluxes, GBA leads to a non-linear (Monod-type) dependence on nutrient concentrations.

Most alternative whole-cell modeling schemes[14–16] are generalizations of FBA and are also based on the optimization of a cellular objective, which is typically set to the cellular growth rate or a proxy thereof. Like GBA, resource balance analysis (RBA)[14] and genome-scale models of metabolism and gene expression (ME)[15] combine a genome-scale metabolic model (as utilized in FBA, Fig. 1) with a protein translation apparatus that converts precursors into protein. While RBA models are formulated at a level of detail typical for FBA models, ME models aim to account comprehensively for all growth-related cellular processes, including, for example, chaperone-assisted protein folding. Contrary to GBA, both methods do not account for metabolite concentrations and assume a linear relationship between fluxes and protein abundances. ME models typically assume constant effective rate constants for reactions, which are set to in vitro[17] or in vivo[18] estimates of turnover numbers ($k_{cat}$). RBA instead uses phenomenological, growth-rate dependent effective kinetic rate constants. These are modeled as linear functions of the growth rate, and parameters are obtained by fitting model-predicted fluxes to proteomics data. Constraint allocation flux balance analysis (CAFBA)[16] is conceptually similar to RBA and ME, but describes the protein costs of biochemical reactions through previously discovered phenomenological growth laws[19,20].

These previous modeling schemes can be considered as approximations to GBA[11] that go beyond FBA by including the protein cost of biochemical fluxes, but that ignore the influence of metabolite concentrations on reaction kinetics and the costs

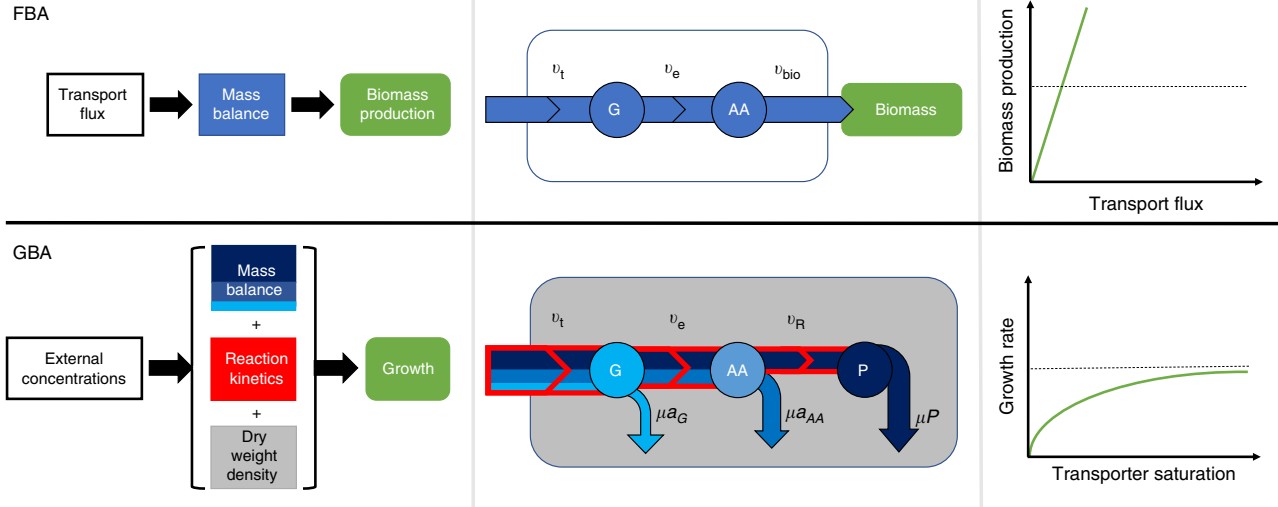

**Fig. 1 A comparison of flux balance analysis (FBA, top) and growth balance analysis (GBA, bottom) for a simple schematic model.** A nutrient G is taken up through a transporter t at rate $v_t$ and is then converted by an enzyme e with rate $v_e$ into a precursor for protein synthesis, AA. In FBA, AA is equated with the biomass, the production of which is maximized while enforcing the stationarity of internal concentrations (blue); this leads to a linear dependence of growth rate on uptake flux. In GBA, AA is converted further into total protein P by a ribosome R, where P represents the sum of the three proteins (t, e, R). GBA maximizes the balanced production of the cellular composition with growth (blue), offsetting the dilution of the cellular components (G, AA, P) with the growth rate $\mu$ indicated by the blue arrows. The reaction fluxes are constrained by non-linear reaction kinetics (red) and a limit on cellular density (dry mass per volume, gray); this leads to a non-linear dependence of growth rate on nutrient concentrations.

incurred through their dilution by growth. Genome-scale implementations of RBA, ME, and CAFBA for model organisms[14–16,21] have been shown to predict some macroscopic phenotypic behavior[14,22]. However, the predicted investment into individual proteins appears to be highly inaccurate, possibly because enzyme kinetics are only treated approximately and metabolite concentrations are not accounted for. Moreover, these methods cannot facilitate a full understanding of phenotypic behavior from basic biochemical and biophysical constraints, and thus do not provide mechanistic insights at the level that would be possible with fully parameterized, genome-scale GBA models.

Due to the central role of kinetic rate laws in GBA, GBA is also closely related to kinetic modeling approaches of cellular metabolism and growth[23,24]. Like GBA, kinetic models implement the mass balance of biochemical reactions while accounting for the dependence of enzyme kinetics on protein and metabolite concentrations. In contrast to GBA and the alternative modeling schemes discussed so far, kinetic modeling approaches do not assume optimality, but simply describe the (steady state) distribution of fluxes and metabolite concentrations resulting from known enzyme concentrations and the kinetic rate laws. However, in vitro kinetic parameters (as reflected in databases such as BRENDA[17]) are very incomplete, and estimates were often made in different experimental settings and are thus not always consistent with each other[23] and with in vivo data[18]. For this reason, enzymatic rate laws in kinetic modeling algorithms are typically parameterized by a fitting procedure that minimizes the discrepancy between model predictions and experimental data (e.g., metabolic fluxes or metabolite concentrations measured across multiple conditions or mutants)[23,24]. Different approaches to kinetic modeling differ from each other in their representation of enzymatic rate laws and in the algorithm used to fit the corresponding parameters. While such fitted parameterizations can lead to accurate predictions of overall cellular physiology, they may show little or no correspondence to experimentally determined kinetic parameters[25]. Moreover, kinetic models typically need to account for substrate-level regulatory interactions to result in realistic predictions[23].

Below, we develop the mathematical foundations for GBA of arbitrarily complex cellular systems. We first describe the constraints that characterize states of balanced growth, and we define elementary growth states (EGSs) by referring to the elementary flux modes (EFMs) of metabolic pathway analysis[26] and FBA. We then show that the reaction fluxes, individual protein concentrations, and growth rate of any EGS are uniquely determined by the set of active reactions and the total cellular protein and individual reactant concentrations. We show how this theoretical framework can be used to understand cellular resource allocation conceptually, and we demonstrate how to analyze specific subsystems for which systems-level effects cancel mathematically.

## Results

**Modeling balanced exponential growth.** Our model assumes that the cell increases exponentially in size, while the concentrations of all cellular components (including the number of membrane constituents per cell volume) remain constant[5]. We do not explicitly model cell division; thus, our model can also be interpreted as describing the growth of a population of cells[8]. In balanced growth, the net production rate of each molecular constituent must balance its dilution by growth, $0 = \frac{dx}{dt}\big|_{\text{production}} - \mu x$, where $x$ denotes the concentration of a given component and $\mu$ is the cellular growth rate[5,8]. The mass conservation in chemical reaction networks is commonly described through a stoichiometric matrix $N$, where rows correspond to metabolites and each column describes the mass balance

of one reaction[26]. Here, we focus on matrices $A$ of active reactions, i.e., $A$ is a sub-matrix of $N$ that contains all columns $j$ for reactions with flux $v_j \neq 0$ and all rows for reactants $i$ involved in these reactions. $A$ also includes a "ribosome" reaction to produce catalytic proteins, encompassing enzymes, transporters, and the ribosome itself. We express concentrations as mass concentrations (mass per volume); accordingly, the entries of $A$ are not stoichiometric coefficients but are mass fractions. The mass conservation of each component can then be stated as

$$A\mathbf{v} = \mu \begin{bmatrix} P \\ \mathbf{a} \end{bmatrix}, \qquad (1)$$

where $\mathbf{v}$ is the flux vector (in units of $[\text{mass}][\text{volume}]^{-1}[\text{time}]^{-1}$), $\mathbf{a}$ is the vector of reactant mass concentrations $a_\alpha$, and $P$ is the sum of the mass concentrations $p_j$ of all proteins $j \in \{1, \ldots, n\}$,

$$P = \sum_j p_j. \qquad (2)$$

The first row of $A$ describes the net production of total protein $P$, which is then distributed among the individual proteins $j$. The remaining rows describe the net production of the reactants $\alpha$.

Each reaction rate $v_j$ is the product of the concentration of its catalyzing protein $p_j$ and a kinetic function $k_j(\mathbf{a})$ that depends on the reactant concentrations $a_\alpha$,

$$v_j = p_j k_j(\mathbf{a}). \qquad (3)$$

We assume that the functional form and kinetic parameters of $k_j(\mathbf{a})$ are known. $k_j(\mathbf{a})$ may depend on the mass concentrations of substrates, products, and other molecules $a_\alpha$ acting as inhibitors or activators, and accounts for the system's thermodynamics. The activity of all reactions $j$ represented in $A$ ($v_j \neq 0$) implies $p_j > 0$ and $k_j(\mathbf{a}) \neq 0$.

Below, we treat the concentrations of total protein $P$ and individual reactants $a_\alpha$ as the state variables of the system, and we show that the fluxes $v_j$, individual protein concentrations $p_j$, and growth rate $\mu$ can be cast as response variables. For a given concentration vector $[P, \mathbf{a}]^T$, we define a balanced growth state (BGS) as a cellular state (characterized by its flux vector $\mathbf{v}$) that satisfies constraints (1), (2) and (3). The set of all such states forms the solution space of balanced growth. On the following pages, we first develop a framework for GBA by characterizing BGSs at a fixed concentration vector $[P, \mathbf{a}]^T$. These characterizations are independent of any physicochemical limits on the concentrations of the cellular components (density constraints); such constraints will, however, become crucial once we examine optimal balanced growth across all feasible concentration vectors. In the main text, we provide an overview over the mathematical structure of GBA and its implications; the formal definitions and theorems are detailed in "Methods", while Supplementary Table 2 lists the symbols used.

**Cellular state defined by the concentration variables.** Let $\mathbf{v}$ be a BGS at concentration vector $\mathbf{y}_0 = [P_0, \mathbf{a}_0]^T$. If we treat $\mathbf{y}_0$ as a constant, then Eq. (1) is mathematically identical to the steady-state constraint fundamental to FBA and to metabolic pathway analysis in general[26]. We call $\mathbf{v}$ an EGS if $\mathbf{v}$ also represents an EFM[27] of the corresponding FBA-type problem defined by the mass-normalized stoichiometric matrix $A$ together with a "biomass reaction" described by $\mathbf{y}_0$ and the flux directions enforced by the signs of the kinetic functions $k_j(\mathbf{a}_0)$ (i.e., $\mathbf{v}$ is a feasible flux vector with minimal support under the FBA-type constraints; "Methods", Definition 3). We can express any BGS as a weighted average of EGSs at the same concentration vector $[P, \mathbf{a}]^T$ (Theorem 3). Moreover, any optimal BGS under a single cellular

density constraint (see below) is also an EGS (Theorem 9 based on refs. [28,29] for EFMs; see also ref. [30]).

Thus, if we characterize the mathematical properties of EGSs, then these properties apply not only to optimal BGSs—which are the main focus of this work—but also to the individual EGS in a decomposition of any BGS. If $A$ is the active stoichiometric matrix of an EGS, it has full column rank (Theorem 4 based on ref. [31]; see also ref. [30]). The full column rank is the only property of EGSs that we will require below. Accordingly, without much loss of generality, we focus on active matrices $A$ that have full column rank for the remainder of this article.

The matrix $A$ may have more rows than columns, in which case some reactant concentrations in $\mathbf{a}$ are linearly dependent on other concentrations[32]. The dependent concentrations $\mathbf{c}$ are not free variables, and hence they can be put aside and dealt with separately. For clarity of presentation, we here present only the case without dependent reactants; the generalization to BGSs with dependent reactants can be treated similarly and is detailed in "Methods".

Without dependent reactants, $A$ is a square matrix with a unique inverse $A^{-1}$, and $\mathbf{x} \equiv [P, \mathbf{a}]^T$ is the corresponding vector of independent concentrations. Multiplying both sides of the mass balance constraint (1) by $A^{-1}$, we obtain (Theorem 5)

$$\mathbf{v} = \mu A^{-1}\mathbf{x}. \qquad (4)$$

The right-hand side of the mass balance constraint (1) quantifies how much of each component $x_i$ needs to be produced to offset the dilution that would otherwise occur through the exponential volume increase at rate $\mu$. $A_{ji}^{-1}$ quantifies the proportion of flux $v_j$ invested into offsetting the dilution of component $i$, and we thus name $A^{-1}$ the investment (or dilution) matrix; see Supplementary Fig. 1 for examples. In contrast to the mass-normalized stoichiometric matrix $A$, which describes local mass balances, $A^{-1}$ describes the structural allocation of reaction fluxes into offsetting the dilution of all downstream cellular components, carrying global, systems-level information.

From the kinetic equation (Eq. (3)), $p_j = v_j/k_j(\mathbf{a})$, and inserting $v_j$ from the investment equation (Eq. (4)) gives

$$p_j = \mu \frac{\sum_i A_{ji}^{-1} x_i}{k_j(\mathbf{a})}. \qquad (5)$$

where $\sum_i$ sums over the total protein and individual reactant concentrations (Theorem 6). Substituting these expressions into the total protein sum (Eq. (2)) and solving for $\mu$ results in the growth equation (Theorem 7)

$$\mu(\mathbf{x}) = \frac{P}{\sum_j \frac{\sum_i A_{ji}^{-1} x_i}{k_j(\mathbf{a})}}. \qquad (6)$$

As detailed in "Methods" (Theorems 5–7), a corresponding result also holds for BGSs with dependent reactants. Thus, for any active matrix $A$ with full column rank (in particular for all active matrices of EGSs) and for any corresponding concentration vector $\mathbf{x}$, there are unique and explicit mathematical solutions for the fluxes $\mathbf{v}$, individual protein concentrations $\mathbf{p}$, and growth rate $\mu$. If $\mu$ (Eq. (6)) and all individual protein concentrations $p_j$ (Eq. (5)) are positive, the cellular state is a BGS; otherwise, no balanced growth is possible at these concentrations.

**Marginal fitness contributions of cellular concentrations.** We now use these relationships to calculate the costs and benefits of concentration changes, which are naturally expressed in terms of relative fitness effects. As above, the main text considers the simpler case without dependent reactants, while the more general

case is treated in "Methods". If fitness is determined predominantly by growth rate[1] (Supplementary Note 1), we can we define the marginal net benefit $\eta_i$ of concentration $x_i$ as the relative change in growth rate[33] due to a small change in $x_i$ ("Methods", Definition 4),

$$\eta_i \equiv \frac{1}{\mu}\frac{\partial \mu}{\partial x_i}; \qquad (7)$$

for example, $\eta_P = \eta_{ATP} = 0.01\,l\,mg^{-1}$ would indicate that an increase of either total protein or ATP concentration by $1\,mg\,l^{-1}$ —if possible—would increase the growth rate by 1%.

To aid in the interpretation of $\eta_i$ below, we define the marginal production cost incurred by the system via protein $j$ as a consequence of increasing concentration $x_i$ at fixed growth rate $\mu$ and kinetics $k_j$,

$$q_i^j \equiv \frac{1}{P}\left(\frac{\partial p_j}{\partial x_i}\right)_{\mu, k_j = \text{const}} = \frac{\mu A_{ji}^{-1}}{P k_j},$$

where the second equality follows from Eq. (5). $q_i^j$ quantifies by how much the concentration $p_j$ of the upstream protein $j$ has to rise in order to offset the increased dilution of the downstream concentration $x_i$. $q_i^j$ is related to the protein control coefficient of metabolic control analysis (MCA); see Supplementary Note 3 for a more detailed summary of the relationship between GBA and MCA[34–36].

Taking the partial derivatives of the growth equation (Eq. (6)) with respect to $P$ and the concentration $a_\alpha$ of reactant $\alpha$, respectively, we find that the marginal net benefits according to Eq. (7) can be expressed as (Theorem 8)

$$\eta_P = \frac{1}{P} - \sum_j q_P^j$$

and

$$\eta_\alpha = \sum_j (u_\alpha^j - q_\alpha^j),$$

with

$$u_\alpha^j \equiv -\frac{1}{P}\left(\frac{\partial p_j}{\partial a_\alpha}\right)_{v_j = \text{const}} = \frac{p_j}{P}\frac{1}{k_j}\frac{\partial k_j}{\partial a_\alpha},$$

where the last equation is derived using $p_j = v_j/k_j$. $u_\alpha^j$ can be interpreted as the marginal kinetic benefit[37] of reactant $\alpha$ to reaction $j$ and quantifies the proportion of protein $p_j$ "saved" due to the change in kinetics associated with an increase in $a_\alpha$. The kinetic benefit $u_\alpha^j$ is a strictly local effect, as it is zero if $a_\alpha$ does not influence the kinetic function $k_j(\mathbf{a})$; we expect $u_\alpha^j$ to be positive if $\alpha$ is a substrate and negative if $\alpha$ is a product of reaction $j$. $u_\alpha^j$ relates directly to the elasticity coefficients of MCA (Supplementary Note 3). Because fluxes are proportional to the concentrations of the catalyzing proteins, the marginal kinetic benefit of total protein is simply $1/P$. Expressions that additionally account for dependent reactants are provided in "Methods".

As seen from the derivation in "Methods", applying the chain rule of differentiation to the growth equation (Eq. (6)) further provides a simple interpretation of the net benefit of component $i$ via reaction $j$ (see "Marginal fitness benefits and costs" in "Methods"; note that because here we assume that there are no dependent reactants, direct and total net benefits as defined in "Methods" are identical). The derivation shows that the marginal net benefit is identical to the reduction of the proteome fractions

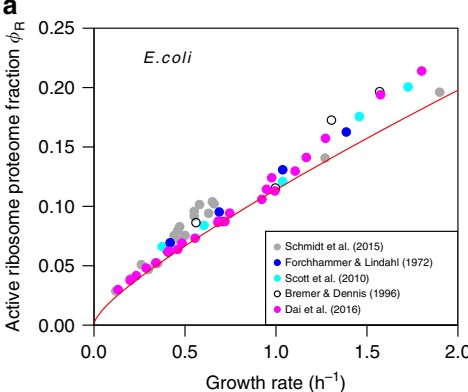
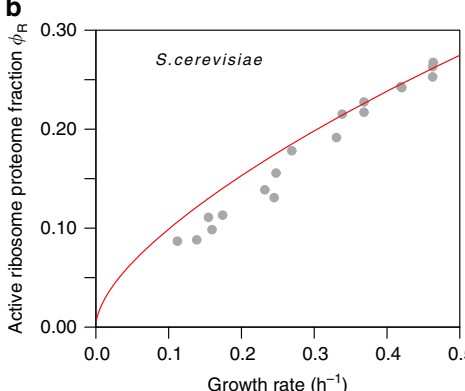

**Fig. 2 GBA predictions of active ribosomal proteome fractions agree with experimental estimates.** Comparison of GBA predictions (red lines, no free parameters) and data. **a** Ribosomal proteome fractions for *E. coli* across different growth conditions, estimated from quantitative proteomics[45] and total RNA/protein ratios[19,42,46,66] ($N = 58$; Pearson's correlation coefficient between observed and predicted values $r^2 = 0.97$, $P < 10^{-43}$; coefficient of determination $R^2 = 0.91$, i.e., the variance of the residuals is only 9% of the variance of the raw data). **b** Ribosomal proteome fractions for *S. cerevisiae* across different growth conditions from quantitative proteomics[47] ($N = 18$; $r^2 = 0.98$, $P < 10^{-14}$; $R^2 = 0.89$).

$\phi_j \equiv p_j/P$ facilitated by the increase in $x_i$ at constant $\mu$,

$$\eta_i = -\sum_j \left( \frac{\partial \phi_j}{\partial x_i} \right)_{\mu = \text{const}}. \tag{8}$$

Thus, for a positive $\eta_i$ and keeping the growth rate $\mu$ constant, a small increase in $x_i$ by $\Delta x_i$ results in a corresponding reduction of the total proteome fraction, $\Sigma_j \Delta \phi_j = -\eta_i \Delta x_i$: at least some proteins are now required at lower concentrations. This result provides a formal justification for the widely held notion that cellular costs lie predominantly in protein production[3,5–9,14,15,19,20,37,38].

**Optimal growth and the balance of marginal net benefits.** Up to this point, we kept $\mathbf{x} = [P, \mathbf{a}]^T$ fixed. We will now characterize optimal growth states, i.e., BGSs with maximal growth rate across all allowed concentration vectors $\mathbf{x}$. To make this problem well defined, we need to consider an additional constraint that reflects the cellular requirement for a minimal amount of free water to facilitate diffusion[39,40]. We implement this constraint by assuming that cellular dry weight per volume is limited to a maximal density $\rho$, where $\rho$ is determined by external osmolarity[40,41] but is otherwise constant across growth conditions[42–44],

$$\rho \geq P + \sum_\alpha a_\alpha. \tag{9}$$

A BGS is a density-constrained BGS (dBGS) if it additionally satisfies constraint (9). At maximal growth rate, the cellular components will utilize the full cellular limit on density to saturate enzymes with their substrates, and thus the inequality in Eq. (9) becomes an equality.

The maximal balanced growth rate $\mu^*$ will be a function of $\rho$. In analogy to the marginal net benefits of cellular components, we define the marginal benefit of the cellular density as the relative fitness increase facilitated by a small increase in $\rho$,

$$\eta_\rho \equiv \frac{1}{\mu^*} \frac{d\mu^*}{d\rho}. $$

Using the method of Lagrange multipliers with the growth equation (Eq. (6)) as the objective function, we derive necessary conditions at optimal growth, which we term balance equations:

$$\forall i \in \{P, \alpha\} : \eta_i = \eta_\rho \tag{10}$$

(Theorem 10). Again, the presentation here assumes that there are no dependent reactants, while a corresponding result is

derived for the general case with dependent reactants in "Methods" ("Optimal density-constrained balanced growth states"). Both with and without dependent reactants, the optimal state is perfectly balanced: the marginal net benefits of all independent cellular concentrations $x_i$ are identical. Thus, if the dry weight density $\rho$ could increase by a small amount (such as 1 mg l$^{-1}$), then the marginal fitness gain that could be achieved by increasing protein concentration by this amount is identical to that achieved by instead increasing the concentration of any reactant $\alpha$ by the same amount. This should not be surprising: if the marginal net benefit of concentration $x_i$ was higher than that of $x_{i'}$, growth could be accelerated by increasing $x_i$ at the expense of $x_{i'}$.

Equation (10) together with Eq. (9) describes a system of $n + 1$ equations for $n + 1$ unknowns, the independent concentrations $x_i$. In realistic cellular systems, this set of equations has a finite number of discrete solutions. Thus, growth rate optimization can be replaced by searching for the solution of the balance equations. If the optimization problem is convex, the conditions given by Eq. (10) are necessary and sufficient, and the solution is unique.

**Quantitative predictions.** If a substrate $\alpha'$ is consumed only by a single reaction that is the only one producing a product $i'$ (with $i' \in \{P, \alpha\}$), the non-local dilution terms in the balance equation ($\eta_{\alpha'} = \eta_{i'}$) cancel, and we are left with a local problem for which only the production cost of $x_{i'}$ and the kinetic benefits of $a_{\alpha'}$ and $x_{i'}$ must be considered. This is the case for protein production in simplified models[38] where the ribosome (R) produces proteins from a single substrate, a generic ternary complex (T). In such models, we can calculate the optimal proteome fraction of actively translating ribosomes, $\phi_R \equiv p_R/P$, from the balance equation $\eta_T = \eta_P$ (Eq. (10)) and its generalization in Theorem 10. The predictions agree quantitatively with experimental values in *E. coli*[45,46] and the yeast *Saccharomyces cerevisiae*[47] across a wide range of growth conditions (Fig. 2).

In contrast to previous approaches based on the analysis of schematic, linear cell models with 2–3 reactions and largely arbitrary kinetic parameters[6–9], our predictions of the scaling of active ribosome fractions with growth rate (Supplementary Fig. 2) are both quantitative and general, as they rely only on the known stoichiometries and kinetics of the ribosome reactions themselves and are independent of any particular network structure. An approximation that ignores the dilution of intermediates and hence the associated production costs ($q_\alpha^j \approx 0$) results in less

accurate predictions of *E. coli* ribosome concentrations especially at high growth rates (Supplementary Fig. 4). In contrast, these approximate predictions are close to observed values for growth on minimal media ($\mu < 1\,h^{-1}$), indicating that the dilution of intermediates, $\mu a_\alpha$, becomes less important at lower growth rates. The latter observation may explain why the relationship between the concentrations of a substrate and its catalysts is well approximated in this regime by simply minimizing their combined mass concentration while keeping the reaction rate constant[48], as this is mathematically equivalent to ignoring the dilution of intermediates.

To obtain a rough quantitative estimate of the marginal net benefits $\eta_i$, we here consider the simplest model of a complete cell, consisting of only a transport protein and the ribosome[3,7] (Supplementary Fig. 2). This model is structurally very similar to previously analyzed schematic whole-cell models. However, contrary to previous models that assumed a fixed total protein concentration as the only density constraint[3,5–9,19], our model's density constraint (9) limits the joint mass concentration of proteins and reactants. Based on the experimentally observed proteome fraction of total dry weight in *E. coli*, $P/\rho = 0.55$[49], we estimate $\frac{\rho}{\mu}\frac{d\mu}{d\rho} = \rho\eta_\rho = 0.66$ ("Methods", Eq. (42)). Thus, a decrease in cellular dry weight density $\rho$ of 1% would lead to a 0.66% reduction in growth rate, emphasizing the biological significance of the density constraint and potentially explaining why *E. coli*'s dry mass density appears to be roughly constant across conditions[42–44].

The cellular density $\rho$ changes when external osmolarity is modified[40]. $\rho\eta_\rho = \frac{d\ln\mu}{d\ln\rho}$ is the slope of the log-log-scale plot of $\mu$ vs. $\rho$ across different external osmolarities. While increases in $\rho$ may have strong effects on diffusion and thus on enzyme kinetics, reductions in $\rho$ due to decreased external osmolarity are within the scope of our model. The very limited available experimental data (three data points from ref. [50], Supplementary Fig. 3) suggest $\rho\eta_\rho \approx 0.66$, the same as our rough estimate from the minimal cell model. An otherwise identical model that limits total protein density[3,5–9,19] $P$ instead of dry mass density predicts a much weaker dependency of growth rate on osmolarity, with $P\eta_P = 0.36$ ("Methods").

## Discussion

At the heart of our mathematical derivations is $A^{-1}$, the inverse of the mass-normalized active stoichiometric matrix $A$ of any given EGS (or, more generally, any given BGS with linearly independent reactions). $A^{-1}$ provides important information on the cellular efficiency. As seen from Eq. (4), $A_{ji}^{-1}$ quantifies which proportion of reaction flux $v_j$ is required to offset the dilution of the downstream cellular component $i$ (either total protein $P$ or reactant $\alpha$). These non-local, structural mass-balance constraints lead to an explicit dependence of reaction fluxes on the cellular concentrations (Eq. (4), Theorem 5). Independently of this, fluxes also depend on concentrations through reaction kinetics (constraint (3)). Combining these two relationships leads to explicit expressions for the individual protein concentrations $p_j$ and for the growth rate $\mu$, casting them as functions of the concentrations $\mathbf{x} = [P, \mathbf{a}]^T$. Accordingly, $A^{-1}$ accounts for all systems-level contributions to the marginal costs and benefits of cellular concentrations $x_i$, while the kinetic functions $k_j(\mathbf{a})$ account for local effects. The insight that optimal, density-constrained states of balanced growth are EGS allowed us to derive the balance equations (Eq. (10)); furthermore, as any BGS can be expressed as a weighted average of EGSs (Theorem 3), our results allow a general characterization of the solution space of balanced growth.

While computational limitations restricted previous studies of balanced growth to specific models with 2–7 reactions, we here provide general results for arbitrarily complex cellular systems. Except for the maximal cellular dry weight density constraint (9), the balanced growth model proposed by Molenaar et al.[5] and utilized subsequently for the analysis of schematic models[3,6–10] is based on assumptions identical to those made for GBA, constraints (1), (2) and (3). Previous authors (with the exception of Faizi et al.[10]) assumed a limit on total protein ("macromolecular") concentrations, while we assume a joint limitation of all cellular solutes (Eq. (9)). The latter choice is justified by the approximate constancy of the cellular dry mass density across growth conditions[42–44], and by an observed relationship between enzyme and substrate concentrations that is consistent with natural selection on the parsimonious use of a limited dry mass density[48].

To make the presentation concise, our development of GBA assumes (i) that all proteins contribute to growth by acting as catalysts or transporters; (ii) that there is a 1-to-1 correspondence between proteins and reactions; (iii) that proteins are not used as reactants; (iv) that all catalysts are proteins; and (v) that cells are optimized for growth. Supplementary Note 2 outlines how to remove these simplifications.

Due to the explicit inclusion of the major physicochemical constraints on cellular growth, GBA models promise to provide a mechanistic understanding of microbial resource allocation and physiology at a depth not achievable with alternative optimization-based models. In principle, exploitation of the balance equations (Eq. (10)) may allow the numerical optimization of cellular systems of realistic size, encompassing hundreds of protein and reactant species. However, several challenges must be overcome before GBA models can be used to make detailed quantitative predictions of genome-scale resource investment and physiology.

The first challenge is the identification of the set of active reactions in a given cellular state, leading to the active stoichiometric matrix $A$. The optimal state is an EFM of the linearized problem (Theorem 9), and thus a direct way to achieve this would be to compute all EFMs of the full stoichiometric matrix compatible with balanced growth (i.e., all support-minimal subsets of reactions that are capable of producing their own reactants plus protein), to apply GBA to each of them, and to then select the EFM resulting in the highest growth rate. While this approach works well for small, schematic models as those in refs. [3,5–9] and may be feasible for coarse-grained models with a few dozen reactions, the number of biomass-producing EFMs in genome-scale networks is too large for them to be calculated exhaustively on current computers[51]. As an approximate alternative, one could restrict this analysis to a subset of candidate EFMs, e.g., based on FBA with molecular crowding[52] and on parsimonious FBA[53] (where fluxes could be scaled by the maximal enzyme turnover rates, $k_{cat}$) or chosen to represent known physiological states (e.g., yield-maximizing vs. overflow metabolism[54]), or one might analyze only EFMs with a pre-specified maximal number of reactions[51].

A further obstacle to the accurate formulation of GBA models is the current incompleteness of knowledge on the kinetic rate laws and parameters needed for the functions $k_j(\mathbf{a})$, the same problem which hampers large-scale kinetic modeling applications[23,24,55] and (with respect to the effective turnover numbers) RBA[14] and ME[15,22] models. Recent developments of high-throughput assays for their estimation from -omics data have led to promising results[14,18,56], suggesting that such approaches may lead to a comprehensive kinetic characterization of model microbes in the future. In parallel, methods from artificial intelligence have been shown to predict enzyme kinetic parameters with reasonable accuracy[57,58], suggesting that these

approaches can augment incomplete in vitro or in vivo parameter sets. Parameter balancing[59] could aid in the completion of a given set of kinetic constants by exploiting the thermodynamic dependencies among biochemical quantities[37]. In addition, GBA parameterizations could be completed similarly to the parameterization of kinetic models[23,24], by fitting model predictions to experimental data acquired across growth conditions. Experience with kinetic models indicates that high predictive power can frequently be achieved with large uncertainties in the parameter sets[23,25], suggesting that even approximate GBA parameterizations may already lead to valuable insights. Finally, finding the optimal state of a genome-scale GBA model requires the numerical solution of a large non-linear optimization problem. The system of $n+1$ equations provided by Eqs. (9) and (10) represents the necessary conditions for optimal growth, and these are important ingredients for developing efficient algorithms to solve the problem.

Although explicit, genome-scale GBA models are built on the same kinetic rate laws as kinetic modeling approaches, their optimization-based methodology does not require enzyme concentrations as model inputs and will likely be more robust to inaccurate kinetic representations. Importantly, GBA will also be much more robust to the omission of regulatory effects of reactants, as these result in additional protein costs but will in most cases have no major influence on the predicted fluxes. On the other hand, kinetic models can be used to assess the cellular response to genetic or environmental perturbations and can utilize mutant data for their parameterization. This is not possible with optimization-based models such as GBA, as they assume that cellular resource allocation in the modeled state is optimal with respect to a known objective function, the balanced growth rate in the case of GBA.

While several challenges have to be met before GBA can be applied to genome-scale balanced growth models, the present work establishes a comprehensive formal basis for such applications. Importantly, this mathematical framework can immediately be applied to the systematic analysis of schematic models, such as those examined in earlier work using numerical methods[5,10] or ad-hoc analytical optimizations[3,6–9]. Moreover, the analytic formulations developed here facilitate the straight forward application of GBA to coarse-grained cellular models of increasing complexity, parameterized from experimental data[19,20,60].

Independent of model details and parameterizations, our mathematical analysis provides general quantitative insights into cellular resource allocation and physiology in states of balanced growth. For example, while previous work has emphasized the central role of proteins in the cellular economy[3,5–9,14,15,19,20,37,38], Eq. (8) provides a rigorous formal justification for this notion in the context of balanced growth. At the same time, whereas the total protein mass concentration $P$ is much higher than the mass concentration of any other cellular constituent $a_\alpha$ in most biological systems, the balance equations show that their marginal net benefits are in fact equal at optimal growth.

The application and further development of the GBA theory may foster an enhanced theoretical understanding of how physico-chemical constraints determine the fitness costs and benefits of cellular organization. Moreover, the explicit expressions for the marginal fitness costs and benefits of cellular concentrations provide a rigorous framework for a quantitative analysis of the cellular economy. We anticipate that this approach will prove fruitful not only in the interpretation of natural and laboratory evolution, but also in optimizing the design of synthetic biological systems.

## Methods

**Overview**. In the first four sections of "Methods", we provide a formal description of Growth Balance Analysis (GBA), detailing the formal definitions, theorems, and proofs that form the basis of the main text. For simplicity of notation, we use the following conventions: $\{\alpha\}$ is the set of all reactants in the active stoichiometric matrix $A$, and $\Sigma_\alpha$ indicates that we sum over all $\alpha \in \{\alpha\}$. We use corresponding notations for the sets of independent basis reactants $\{\beta\}$, with concentrations $b_\beta$, and dependent reactants $\{\gamma\}$, with concentrations $c_\gamma$ (see below). As explained in Definition 1 below, stoichiometric matrices are always in units of mass fractions, not stoichiometric coefficients. The last two sections describe the calculations of the optimal ribosome proteome fractions and the dependence of maximal growth rate on cellular water content.

**Characterization of balanced growth states**. First, we introduce the fundamental definitions that characterize the solution space of balanced cellular growth. We define BGSs and generalize the concept of EFMs from linear constraint-based models to EGSs (defined as flux vectors). We then introduce several theorems on the characterization and decomposition of BGSs.

In the formulation presented here, we assume that proteins do only act as catalysts and not as substrates of reactions. Hence, neither total protein nor individual proteins are considered "reactants".

Definition 1 (BGSs): Let $\mathbf{v}' \in \mathbb{R}^{n'}$ be the vector of fluxes through the biochemical reactions that occur in a cell, in units of [mass][volume]$^{-1}$[time]$^{-1}$. Let $\mathbf{v} \in \mathbb{R}^n_{\neq 0}$, $n \leq n'$, be the subvector of $\mathbf{v}'$ that contains all active fluxes of $\mathbf{v}'$ (i.e., all entries $v'_k \neq 0$). Let $\mathbf{y} \equiv [P, \mathbf{a}]^T \in \mathbb{R}^{m+1}_{>0}$ be a corresponding vector of total protein concentration $P$ and individual reactant concentrations $a_\alpha$, $\alpha \in \{1, ..., m\}$, where each $a_\alpha$ is consumed or produced by at least one of the fluxes $v_i$; $\mathbf{y}$ is in units of [mass][volume]$^{-1}$. Let $A \in \mathbb{R}^{(m+1) \times n}$ be the corresponding active stoichiometric matrix in mass fraction units, i.e., column $j$ of $A$ describes reaction $j$ with flux $v_j$, row $i$ of $A$ corresponds to the cellular component $y_i$, and each column is mass balanced. Thus, the sum of negative entries in each column is $S_- = -1$ and the sum of positive entries of each column is $S_+ = +1$; for reactions that involve an external substrate not represented by a row of $A$, $-1 < S_- \leq 0$, while for reactions that involve an external product, $0 \leq S_+ < 1$.

Let $\mathbf{p} \in \mathbb{R}^n_{>0}$ be the vector of individual protein concentrations (in units of [mass][volume]$-1$), where protein $j$ catalyzes reaction $j$; for simplicity, we assume that the "ribosome" catalyzing protein production is also itself a protein (but see Supplementary Note 2 for how to remove this simplification). Let $\mathbf{k}(\mathbf{a})$ be a vector of kinetic functions, $\mathbf{k}: \mathbb{R}^m_{>0} \mapsto \mathbb{R}^n_{\neq 0}$, where $k_j(\mathbf{a})$ is in units of [time]$^{-1}$.

Then $\mathbf{v}$ is a balanced growth state (BGS) at growth rate $\mu$ if and only if it fulfills the following three constraints:

$$A\mathbf{v} = \mu \begin{bmatrix} P \\ \mathbf{a} \end{bmatrix} \tag{11}$$

$$v_j = p_j k_j(\mathbf{a}) \tag{12}$$

$$P = \sum_j p_j. \tag{13}$$

A BGS $\mathbf{v}$ at growth rate $\mu$ is a density-constrained BGS (dBGS) if it additionally fulfills the constraint on total dry mass density

$$\rho \geq P + \sum_\alpha a_\alpha. \tag{14}$$

Constraint (11) implements mass balance, constraint (12) implements concentration-dependent reaction kinetics, while constraint (13) implements a constraint on the total proteome concentration. The kinetic constraint (12) assumes that the flux through each reaction is linear in the concentration of the catalyzing enzyme, while the dependence on the reactant concentrations $a_\alpha$ will typically be non-linear. For simplicity of notation, we will sometimes make the dependence of kinetics on $\mathbf{a}$ implicit, i.e., we will use $k_j \equiv k_j(\mathbf{a})$.

In the above definitions, we define a BGSs (or dBGSs) as a function of the set of active reactions (corresponding to the columns of $A$) and the concentration vector $\mathbf{y} = [P, \mathbf{a}]^T$. For a given active stoichiometric matrix $A$, the set of all such states at all concentrations $\mathbf{y} \in \mathbb{R}^{m+1}_{>0}$ defines the solution space of balanced growth (or of density-constrained balanced growth if only concentrations $\mathbf{y}$ that respect constraint (14) are considered).

Based on biophysical considerations, we might replace Eq. (14) with separate density constraints on the total volume concentration inside each cellular compartment[39] and on the total area occupied by non-lipid membrane components per membrane area[5,61]. An even simpler density constraint imposed in most previous models[3,5–9,14,15] is to fix total protein concentration $P$ to a constant value. However, it has been shown that $P$ decreases with increasing growth rate, whereas total dry mass density is approximately constant across conditions[42–44]. Thus, while a constant $P$ allows to simplify the presentation, Eq. (9) provides a biologically more meaningful constraint; moreover, this constraint allows us to determine the costs and benefits of varying the total protein concentration.

De Groot et al. have defined BGSs for a similar problem[30]. In their formulation, the dimensions of the concentration vector $\mathbf{y}$ include not only total protein $P$, but all individual protein concentrations $p_j$. This more general problem formulation

comes at the cost of more involved decomposition rules[30] compared with Theorem 2.

We now provide the basis for linking BGSs to EFMs, which are defined for FBA-type linear constraint-based problems[27] and which have been extended to proteome-constrained models[28,29].

Definition 2 (EFMs): Let $\mathbf{v} \in \mathbb{R}^n$, $\mathbf{y} = [P, \mathbf{a}]^T \in \mathbb{R}_{>0}^{m+1}$, and $A \in \mathbb{R}^{(m+1) \times n}$ be as in Definition 1. Let $\mathbf{k}^{(\text{eff})} \in \mathbb{R}_{\neq 0}^n$ be a vector of effective kinetic constants. Then we call $\mathbf{v}$ a feasible flux vector at biomass production rate $v_{\text{bio}}$ if and only if it fulfills the following constraints:

$$A\mathbf{v} = v_{\text{bio}} \begin{bmatrix} P \\ \mathbf{a} \end{bmatrix} \tag{15}$$

$$v_j \leq p_j k_j^{(\text{eff})} \tag{16}$$

$$P = \sum_j p_j. \tag{17}$$

A feasible flux vector $\mathbf{v}$ is a representative of an elementary flux mode (EFM) if and only if it is non-decomposable, i.e., it fulfills the following additional constraint[27]: There exists no couple of feasible flux vectors $\mathbf{v}'$, $\mathbf{v}''$ such that $\mathbf{v} = \lambda_1 \mathbf{v}' + \lambda_2 \mathbf{v}''$ with $\lambda_1, \lambda_2 > 0$ and where both $\mathbf{v}'$ and $\mathbf{v}''$ have at least the same number of zeroes as $\mathbf{v}$, while at least one of them contains more zeroes than $\mathbf{v}$.

If we consider the concentration vector $\mathbf{y} = [P, \mathbf{a}]^T$ as a descriptor of a constant biomass composition, we see that constraint (15) is mathematically equivalent to the standard steady-state constraint of FBA and metabolic pathway analysis[26] problems, formulated without an artificial "biomass reaction" in $A$ (see, for example, Eq. (2) in ref. [62]). Note that in the definition of EFMs, both the biomass composition $\mathbf{y} = [P, \mathbf{a}]^T$ and the effective kinetics $\mathbf{k}^{(\text{eff})}$ are assumed to be constant; thus, the constraints (15)–(17) that define the space of feasible flux vectors are fully linear. In contrast, constraint (12) in Definition 1 defines reaction kinetics as a function of the reactant concentrations $\mathbf{a}$.

Definition 3 (EGS): A BGS $\mathbf{v}$ at concentrations $\mathbf{y} = [P, \mathbf{a}]^T$ is an elementary growth state (EGS) if and only if it is a representative of a corresponding EFM, i.e., $\mathbf{v}$ represents an EFM of the corresponding linear problem with constant biomass $\mathbf{y}$ and effective kinetic constants $\mathbf{k}^{(\text{eff})} = \mathbf{k}(\mathbf{a})$.

We emphasize that $\mathbf{v}$ is an EFM of the corresponding linearized (FBA-like) problem (see Definition 2), not of the balanced growth problem (Definition 1) from which it is derived. EFMs are defined as equivalence classes of minimal feasible steady-state flux distributions, whose members can be converted into each other by multiplication with a positive scalar[27]. This definition cannot be generalized to balanced growth states, as multiples of a feasible flux vector generally do not satisfy constraint (11). For this reason, de Groot et al. have generalized the concept of EFMs to equivalence classes of minimal sets of active reactions in BGSs, termed elementary growth modes (EGMs)[30].

Theorem 1 (Existence of solutions): Let $\mathbf{y} = [P, \mathbf{a}]^T$ be a concentration vector and $\mu > 0$ be a growth rate. For any flux vector $\mathbf{v}'$ that satisfies the mass balance constraint (11), there exists a unique BGS $\mathbf{v} = \lambda \mathbf{v}'$ with $\lambda > 0$ if all fluxes run in the direction compatible with the reaction kinetics (i.e., $\forall j: k_j v_j > 0$), and no such BGS otherwise.

Proof: From constraint (12), it is clear that if $k_j v_j \leq 0$, no BGS with $p_j > 0$ exists. For $k_j \neq 0$, the concentration of protein $j$ is uniquely defined by $p_j = v_j / k_j$ (constraint (12)). Let $P' = \sum_j v'_j / k_j$ be the total protein concentration associated with $\mathbf{v}'$. Then setting $\lambda \equiv P / P'$ results in the only flux vector that fulfills all constraints of Definition 1. □

Next, we use this result to show that any weighted average of BGSs is itself a BGS.

Theorem 2 (A weighted average of BGSs is a BGS): Let $(\mathbf{v}^{(1)}, \ldots, \mathbf{v}^{(k)})$ be an ordered set of BGSs for the concentration vector $\mathbf{y} = [P, \mathbf{a}]^T$ with growth rates $(\mu^{(1)}, \ldots, \mu^{(k)})$, but with potentially different active stoichiometric matrices $A^{(l)}$. Let $A$ be the stoichiometric matrix that combines all reactions represented in $(A^{(1)}, \ldots, A^{(k)})$, i.e., the columns of $A$ consist of all unique columns of $(A^{(1)}, \ldots, A^{(k)})$. Let $(\mathbf{v}'^{(1)}, \ldots, \mathbf{v}'^{(k)})$ be a representation of the individual BGSs $\mathbf{v}^{(l)}$ in the flux space defined by $A$, i.e., $v'_j^{(l)} = 0$ for all columns (reactions) of $A$ not represented in $A^{(l)}$. Then any weighted average $\mathbf{v} = \sum_l w_l \mathbf{v}'^{(l)}$ of these extended flux vectors (with weights $w_l > 0$ and $\sum_l w_l = 1$) is itself a BGS for $\mathbf{y}$, with a growth rate that is the weighted average of the individual growth rates, $\mu = \sum_l w_l \mu^{(l)}$.

Proof: The mass balance constraint (11) is linear in the fluxes and growth rates, and is hence also fulfilled for the weighted averages. The protein concentrations of each BGS $\mathbf{v}'^{(l)}$ are $p'_j^{(l)} = v'_j^{(l)} / k_j$. To satisfy the reaction kinetics constraint (12), the protein concentrations of the weighted average are $p_j = v_j / k_j = \sum_l w_l v'_j^{(l)} / k_j = \sum_l w_l p'_j^{(l)}$. As each BGS ($l$) fulfills the proteome constraint (13), $\sum_j p_j = \sum_j \sum_l w_l p'_j^{(l)} = \sum_l w_l P = P$, and thus $\mathbf{v}$ is a BGS. □

We can now use Theorems 1 and 2 together with results on EFMs to show that any BGS can be decomposed into a weighted average of EGSs.

Theorem 3 (BGSs are weighted averages of EGSs): Any BGS $\mathbf{v}$ for the concentration vector $\mathbf{y} = [P, \mathbf{a}]^T$ can be decomposed into a weighted average of EGSs at $\mathbf{y}$.

Proof: $\mathbf{v}$ is a feasible flux vector for the linearized problem defined by constraints (15)–(17) at constant biomass $\mathbf{y}$. The direction of reaction $j$ is fixed by the sign of $k_j^{(\text{eff})} = k_j(\mathbf{a})$, i.e., all reactions are irreversible. Under these conditions, it has been shown that $\mathbf{v}$ is a convex combination of EFMs $\mathbf{v}'^{(l)}$ of the linear problem[27], i.e., $\mathbf{v} = \sum_l w'_l \mathbf{v}'^{(l)}$ with $w'_l > 0$. From Theorem 1, we know that for each of these EFMs, there exists a unique BGS $\mathbf{v}^{(l)} = \lambda_l \mathbf{v}'^{(l)}$ with $\lambda_l > 0$; according to Definition 3, this is an EGS. Thus, we can write $\mathbf{v} = \sum_l w_l \mathbf{v}^{(l)}$ as a linear combination of EGSs, with weights $w_l \equiv w'_l / \lambda_l$.

To prove that $\mathbf{v}$ is a weighted average of the $\mathbf{v}^{(l)}$, it remains to be shown that $W \equiv \sum_l w_l = 1$. According to Theorem 2, a weighted average $\mathbf{v}'' \equiv \sum_l \frac{w_l}{W} \mathbf{v}^{(l)} = \frac{1}{W}\mathbf{v}$ will also be a BGS. However, Theorem 1 states that there exists only one BGS in the direction of $\mathbf{v}$, and thus $W = 1$. □

**Growth equations.** In this section, we assume that the concentrations of total protein and of individual reactants, $\mathbf{y} \equiv [P, \mathbf{a}]$ are known. Mass conservation (constraint (11)) and reaction kinetics (constraint (12)) relate reaction fluxes to the concentration vector in two fundamentally different ways. We will now exploit this fact to eliminate the flux variables and to derive explicit expressions for $\mathbf{v}$, $\mathbf{p}$, and $\mu$.

Note that because the concentrations $\mathbf{y}$ are used as state variables in these analyses, no explicit consideration of constraints on cellular density, such as constraint (14), is necessary. The given concentrations $\mathbf{y}$ may obey constraint (11) or alternative density constraints, such as independent constraints on the density of cellular compartments, but these will not be used here. They will only become important when we vary $\mathbf{y}$ to find states of maximal growth rate in a later section.

An important requirement for the analyses below is that the active stoichiometric matrix $A$ has full column rank, motivating the next theorem.

Theorem 4 (The active reactions of an EGS are linearly independent): Let $A \in \mathbb{R}^{(m+1) \times n}$ be the active stoichiometric matrix of an EGS. Then A has full column rank $n$, i.e., the columns of A are linearly independent.

Proof: According to the definition of EGSs (Definition 3), $A$ is also the active matrix of the corresponding linearized (flux balance type) problem. It has previously been shown[31] that the active stoichiometric matrix $A$ of an EFM of a linear flux-balance problem has full column rank if $A$ is formulated without an explicit "biomass" reaction (as in Definition 2). □

According to this theorem, the following theorems—which assume that $A$ has full column rank—can in particular be applied to EGSs (and, as we will see below in Theorem 9, thus also to dBGSs with maximal growth rate).

Theorem 5 (Investment equation): Let $A \in \mathbb{R}^{(m+1) \times n}$ be an active stoichiometric matrix of a flux vector $\mathbf{v}$ that fulfills the mass balance constraint (11) with concentration vector $\mathbf{y} = [P, \mathbf{a}]^T$, where $A$ has full column rank $n$. Then we can split $A$ into two submatrices $B \in \mathbb{R}^{n \times n}$ and $C \in \mathbb{R}^{(m+1-n) \times n}$,

$$A = \begin{bmatrix} B \\ C \end{bmatrix},$$

such that $B$ is a non-singular (invertible) square matrix and each row of $C$ is a linear combination of rows of $B$. Let $B^{-1}$ be the inverse of $B$. Let $\mathbf{b}$ be the subvector of reactant concentrations $\mathbf{a}$ that correspond to the rows of $B$, $\mathbf{c}$ be the subvector of the reactant concentrations that correspond to the rows of $C$, and $\mathbf{x} \equiv [P, \mathbf{b}]^T$. Then $\mathbf{v}$ is given by

$$\mathbf{v} = \mu B^{-1} \mathbf{x}.$$

The dependent reactant concentrations $\mathbf{c}$ are linear combinations of the independent concentrations $\mathbf{x}$,

$$\mathbf{c} = D\mathbf{x}, \tag{18}$$

with the dependence matrix $D \equiv CB^{-1}$.

Proof: The active stoichiometric matrix $A$ may have more rows than columns. In this case, $m + 1 > n$, and the rows for exactly $n$ metabolites are linearly independent, as row and column rank must equal. As a consequence, the remaining $m + 1 - n$ metabolite concentrations are linearly dependent on the concentrations of the $n$ independent metabolites. These dependent concentrations are not free variables, and hence they can be put aside and dealt with separately.

We decompose the linear system of equations represented by constraint (11) into two parts, rearranging the rows of $A$ into matrices $B$, $C$ such that $B$ contains the rows for the independent reactants. As $A$ has full column rank, choosing linearly independent rows results in a square matrix $B$ of full rank (#rows($B$) = rank($B$) = rank($A$)). Let $\mathbf{b}$ be the subvector of reactant concentrations $\mathbf{a}$ that correspond to the rows of $B$, and let $\mathbf{c}$ be the subvector of the remaining reactant concentrations corresponding to the rows of $C$. We can then split the mass balance constraint (11) into two separate equations:

$$B\mathbf{v} = \mu \begin{bmatrix} P \\ \mathbf{b} \end{bmatrix}$$

$$C\mathbf{v} = \mu \mathbf{c},$$

$B$ is a square matrix of full rank, so there is always a unique inverse $B^{-1}$. Multiplying both sides of the first equation by $B^{-1}$ from the left, we obtain the

desired equation for $\mathbf{v}$. Inserting this result into the second equation results in the desired equation for $\mathbf{c}$. $\square$

Thus, if $A$ has full rank, then any flux vector $\mathbf{v}$ respecting the flux balance constraint (11) is uniquely defined and is a linear combination of the total protein concentration $P$ and the independent metabolite concentrations $\mathbf{b}$. Each entry of the inverse matrix $B_{ji}^{-1}$ quantifies the proportion of flux $j$ invested into the dilution of component $i$, and we thus name $B^{-1}$ the investment (or dilution) matrix (see Supplementary Fig. 1 for examples). In contrast to the stoichiometric matrix $A$, which describes local mass balances (constraint (11)), $B^{-1}$ describes the structural allocation of reaction fluxes into the production of cellular components diluted by growth, and thus carries global, systems-level information.

$B$ corresponds to the reduced stoichiometric matrix in ref. [32]. $D$ describes the linear dependence of the dependent concentrations $\mathbf{c}$ on $P$ and $\mathbf{b}$; it is identical to the link matrix in ref. [32]. The relationship between $A$ and $B$, $C$ can be understood in terms of matroid theory, where the rows of $B$ form a basis for the matroid spanned by the rows of $A$, and the set of rows of $C$ is the closure for the set of rows of $B$. If the choice for the partitioning of $A$ into both $B$ and $C$ is not unique, some partitionings may be pathological and should be avoided (Supplementary Note 4).

When $A$ is not square, $B$ includes a proper subset of the rows in $A$, and thus $B$ on its own is not mass balanced. The "missing" mass fluxes are balancing $\mathbf{c}$, and hence the flux investment into $\mathbf{c}$ is already accounted for by the investment equation in Theorem 5.

We are now in a position to express the individual protein concentrations and the growth rate of a BGS as explicit functions of the concentrations $\mathbf{y} = [P, \mathbf{a}]^T$.

Theorem 6 (Individual protein concentrations as a function of the independent concentrations): Let $A \in \mathbb{R}^{(m+1) \times n}$ be an active stoichiometric matrix with full column rank $n$, and let $\mathbf{x} = [P, \mathbf{b}]^T$ be the independent concentration vector with corresponding index $i \in \{P, \beta\}$. Let $\mathbf{v}$ be a corresponding BGS. Let $B$ and $D$ be the basis and dependency matrices, respectively, as defined in Theorem 5. Then the concentration of the protein catalyzing reaction $j$ is

$$p_j = \mu \frac{\sum_i B_{ji}^{-1} x_i}{k_j(\mathbf{a})}.$$

Proof: As $A$ is an active matrix, all fluxes $v_j = p_j k_j(\mathbf{a})$ (constraint (12)) are non-zero. We can thus express the individual protein concentrations as $p_j = v_j/k_j(\mathbf{a})$. Inserting $v_j$ from the investment equation (Theorem 5) directly leads to the above equation. $\square$

We now insert the equations for the individual proteins into the total protein constraint (13) to obtain an explicit expression for the growth rate.

Theorem 7 (Growth equation): Let $A \in \mathbb{R}^{(m+1) \times n}$ be an active stoichiometric matrix with full column rank $n$, and let $\mathbf{y} = [P, \mathbf{a}]^T$ be a concentration vector. Let $\mathbf{v}$ be a corresponding BGS. Let $B$ and $D$ be the basis and dependency matrices, respectively, as defined in Theorem 5. Then the growth rate is

$$\mu(P, \mathbf{a}) = \frac{P}{\sum_j \frac{\sum_i B_{ji}^{-1} x_i}{k_j(\mathbf{a})}}$$

if for all reactions $\frac{p_j}{\mu} = \frac{\sum_i B_{ji}^{-1} x_i}{k_j(\mathbf{a})} > 0$, and no balanced growth is possible otherwise.

Proof: According to Theorem 6, the individual protein concentrations are $p_j = \mu \frac{\sum_i B_{ji}^{-1} x_i}{k_j(\mathbf{a})}$. The flux $v_j$ catalyzed by protein $j$ must be active, and thus $p_j$ has to be positive for all $j$. Substituting the expressions for $p_j$ into the proteome constraint (13), we obtain

$$P = \mu \sum_j \frac{\sum_i B_{ji}^{-1} x_i}{k_j(\mathbf{a})}.$$

The sum on the r.h.s. is positive, and dividing by it results in the growth equation. $\square$

Thus, if the active matrix $A$ of a BGS is full rank, there are unique and explicit mathematical solutions for $\mathbf{p}$, $\mathbf{v}$, and $\mu$. In particular, this is the case for optimal growth states (Theorem 9), as well as for all other EGSs. In this section, we did not impose any density constraints (such as constraint (14)), and thus Theorems 1–7 remain valid under arbitrary density constraints as long as these are respected by the concentration vector $\mathbf{y} = [P, \mathbf{a}]^T$.

**Marginal fitness benefits and costs.** In this section, we first define marginal fitness benefits and costs of concentrations. As in the previous section, the considerations in this section make no use of the density constraint (14), and thus remain valid under alternative density constraints. After introducing the definitions, we show how to calculate and to interpret the costs and benefits.

Definition 4 (Marginal costs and benefits): Let $\mathbf{v}$ be a BGS with growth rate $\mu$. Let $i \in \{P, \beta\}$ be an index of the independent concentration vector $\mathbf{x} = [P, \mathbf{b}]^T$. Then the direct marginal net benefit of concentration $x_i$ is defined as the relative change in growth rate due to a small change in $x_i$[33],

$$\eta_i^0 \equiv \frac{1}{\mu} \frac{\partial \mu}{\partial x_i}.$$

Analogously, we define the marginal benefit of dependent reactant $\gamma$ as

$$\eta_\gamma^c \equiv \frac{1}{\mu} \frac{\partial \mu}{\partial c_\gamma}. \tag{19}$$

The (total) marginal net benefit of $x_i$ is then defined as the relative change in growth rate due to a small change in $x_i$, accounting for the resulting changes in the concentration of dependent metabolites $c_\gamma$,

$$\eta_i \equiv \frac{1}{\mu} \left( \frac{\partial \mu}{\partial x_i} + \sum_\gamma \frac{\partial \mu}{\partial c_\gamma} \frac{\partial c_\gamma}{\partial x_i} \right) = \eta_i^0 + \sum_\gamma D_{\gamma i} \eta_\gamma^c, \tag{20}$$

where the second equality follows directly from Eq. (18).

A change $\delta x_i$ of $x_i$ ($i \in \{P, \beta\}$) causes a correlated change of each dependent concentration $\delta c_\gamma = D_{\gamma i} \delta x_i$ (Eq. (18)). Thus, a change by $\delta x_i$ results in a total change of the utilization of cellular density by $\kappa_i \delta x_i$, with the density factor defined as

$$\kappa_i \equiv 1 + \sum_\gamma D_{\gamma i}.$$

To help in the interpretation of the marginal net benefits, we will relate them in the next theorem to two explicit definitions of costs and benefits, respectively. The marginal production cost of the cellular concentration $x_i$ is defined as

$$q_i^j \equiv \frac{1}{P} \left( \frac{\partial p_j}{\partial x_i} \right)_{\mu, k_j = \text{const}},$$

where the subscript of the parenthesis indicates which variables are kept constant in the derivative. $q_i^j$ can be interpreted as the additional amount of protein $j$ required to offset the increased dilution of $x_i \in \{P, \beta\}$ at growth rate $\mu$ and fixed kinetics $k_j$. We define the marginal kinetic benefit of the reactant concentration $b_\beta$ as

$$u_\beta^j \equiv -\frac{1}{P} \left( \frac{\partial p_j}{\partial b_\beta} \right)_{v_j = \text{const}},$$

and we make corresponding definitions $u_\gamma^j$ for dependent concentrations $c_\gamma$. The marginal kinetic benefits can be interpreted as the fraction of proteins $j$ saved at constant flux $v_j$ due to the increased saturation of reaction $j$ with reactant $\beta$ or $\gamma$, respectively.

The marginal net benefits can now be expressed as differences between benefits and costs. To calculate the direct marginal net benefits $\eta_i^0$, we must use the growth equation derived in Theorem 7,

$$\mu(P, \mathbf{a}) = \frac{P}{\sum_j \frac{\sum_i B_{ji}^{-1} x_i}{k_j(\mathbf{a})}} = \frac{P}{\sum_j \frac{p_j}{\mu}} = \frac{1}{\sum_j \frac{\phi_j}{\mu}}, \tag{21}$$

where we defined proteome fractions $\phi_j \equiv p_j/P$. The first form given for $\mu(P, \mathbf{a})$ here quantifies growth as a function of the state variables $x_i$, and it would be straight forward to calculate $\eta_i^0$ from this expression. However, to establish a formal link between marginal net benefits and protein investment, we will instead go via the second form, which arises from Theorem 6 and was used to derive the growth equation, and the third form, which expresses this relationship in terms of the proteome fractions $\phi_j$. When we take the partial derivatives with respect to the state variables $x_i$ in the second and third forms, we must make sure that we keep the right terms constant: when expressed in terms of the $x_i$, the expression $p_j/\mu$ is in fact independent of $\mu$ (Theorem 6), and we hence need to take the derivatives while keeping $\mu$ constant. We thus get for the direct marginal net benefits:

$$\eta_i^0 \equiv \frac{1}{\mu} \frac{\partial \mu}{\partial x_i} = \frac{1}{\mu} \left( \frac{\partial}{\partial x_i} \frac{1}{\sum_j \frac{\phi_j}{\mu}} \right)_{\mu = \text{const}} = \left( \frac{\partial}{\partial x_i} \frac{1}{\sum_j \phi_j} \right)_{\mu = \text{const}}$$

$$= -\frac{1}{\left( \sum_j \phi_j \right)^2} \sum_j \left( \frac{\partial \phi_j}{\partial x_i} \right)_{\mu = \text{const}} = -\sum_j \left( \frac{\partial \phi_j}{\partial x_i} \right)_{\mu = \text{const}},$$

where we used the fact that the proteome fractions must add to 1, $\sum_j \phi_j = 1$. Thus, the direct marginal net benefit of the cellular concentration $x_i$ is identical to the total associated changes in proteome fractions caused by this change.

Again looking at Eq. (21), we can further analyze the nature of the proteome changes caused by a change in the cellular concentration $x_i$. Let us first consider a reactant concentration $x_i = b_\beta$. Applying the chain rule of differentiation to $\phi_j = p_j/P = \mu \sum_i B_{ji}^{-1} x_i / k_j(\mathbf{a})$, we have to add the partial derivatives with respect to $x_i = b_\beta$ in the numerator $\mu \sum_i B_{ji}^{-1} x_i = v_j$ (keeping $\mu$ and $k_j$ constant) and in the denominator $k_j(\mathbf{a})$ (keeping the numerator $v_j$ constant, which also guarantees that $\mu$ is constant). Thus, we can write the direct marginal net benefits of the independent

reactant concentration $b_\beta$ in terms of proteome changes as

$$\eta_\beta^0 = -\sum_j \left(\frac{\partial \phi_j}{\partial b_\beta}\right)_{v_j = \mathrm{const}} - \sum_j \left(\frac{\partial \phi_j}{\partial b_\beta}\right)_{\mu, k_j = \mathrm{const}}$$

$$= -\frac{1}{P}\sum_j \left(\frac{\partial p_j}{\partial b_\beta}\right)_{v_j = \mathrm{const}} - \frac{1}{P}\sum_j \left(\frac{\partial p_j}{\partial b_\beta}\right)_{\mu, k_j = \mathrm{const}}$$

$$= \sum_j \left(u_\beta^j - q_\beta^j\right),$$

where in the last line we inserted the definitions of the marginal kinetic benefits and production costs (Definition 4).

Performing an analogous calculation for the direct net benefit of the total protein concentration $P$ (noting that we now need to take the derivative with respect to the numerator of the growth equation but not with respect to $(k_j)$), we obtain

$$\eta_P^0 = -\sum_j \left(\frac{\partial \phi_j}{\partial P}\right)_{\mu = \mathrm{const}} = -\left(\frac{\partial}{\partial P}\frac{1}{P}\sum_j p_j\right)_{\mu = \mathrm{const}}$$

$$= \frac{1}{P^2}\sum_j p_j - \frac{1}{P}\sum_j \left(\frac{\partial p_j}{\partial P}\right)_{\mu = \mathrm{const}}$$

$$= \frac{1}{P} - \sum_j q_P^j,$$

where we used $\sum_j p_j = P$ and where we inserted the definition of the marginal production cost of $P$ (Definition 4, with $k_j$ independent of $P$) in the last line. The positive term $1/P$ in the direct net benefit of total protein quantifies the marginal benefit of increasing the total protein concentration $P$, which accelerates all reactions linearly.

We have thus proven the next Theorem, which elucidates how costs and benefits of cellular compounds are naturally related to protein use; this connection has been proposed before[33,37] but is derived here rigorously from first principles.

Theorem 8 (Direct marginal net benefits): The direct marginal net benefit of any independent cellular concentration $x_i$ ($i \in \{P, \beta\}$) is the negative of the total associated change in relative protein concentrations at constant growth rate $\mu$,

$$\eta_i^0 = -\sum_j \left(\frac{\partial \phi_j}{\partial x_i}\right)_{\mu = \mathrm{const}}. \tag{22}$$

The direct marginal net benefits of the total protein concentration $P$ and of independent reactant concentrations $b_\beta$ ($\beta \in \{1, \ldots, m\}$), respectively, are

$$\eta_P^0 = \frac{1}{P} - \sum_j q_P^j$$

$$\eta_\beta^0 = \sum_j (u_\beta^j - q_\beta^j).$$

The marginal production cost $q_i^j$ is the fraction of extra protein $j$ expended to offset the additional dilution of concentration $x_i$ at rate $\mu$ and fixed saturation $k_j$; it can be calculated from the growth equation (Theorem 7) as

$$q_i^j \equiv \frac{1}{P}\left(\frac{\partial p_j}{\partial x_i}\right)_{\mu, k_j = \mathrm{const}} = \frac{\mu B_{ji}^{-1}}{P k_j}. \tag{23}$$

The marginal kinetic benefit $u_\beta^j$ is the fraction of protein $j$ saved due to its increased saturation with reactant $\beta$; it is calculated from the growth equation as

$$u_\beta^j \equiv -\left(\frac{\partial \phi_j}{\partial b_\beta}\right)_{v_j} = \frac{\phi_j}{k_j}\frac{\partial k_j}{\partial b_\beta}.$$

The marginal kinetic benefits of dependent reactants $\gamma$ are

$$\eta_\gamma^c \equiv \frac{1}{\mu}\frac{\partial \mu}{\partial c_\gamma} = \sum_j u_\gamma^j,$$

where $u_\gamma^j$ is calculated analogously to the marginal kinetic benefits of independent reactants, $u_\beta^j$.

**Optimal density-constrained balanced growth states.** So far, we have considered BGS for a given set of active reactions (corresponding to the columns of $A$) and given concentrations $\mathbf{y} = [P, \mathbf{a}]^T$, where $\mathbf{y}$ may or may not have respected any particular density constraint. We now examine density-constrained BGSs (dBGSs) with maximal growth rate given the set of active reactions, optimized over all concentration vectors $\mathbf{y} = [P, \mathbf{a}]^T \in \mathbb{R}_{>0}^{m+1}$ that respect the density constraint (14). As a preparation for these analyses, we first show that states of optimal growth are EGSs.

**Theorem 9** (dBGSs with maximal growth rate are EGSs). Let $N$ be a stoichiometric matrix of a general balanced growth model. Let $\mathbf{v}^*$ be a dBGS that maximizes the growth rate of the general problem. Then $\mathbf{v}^*$ is an EGS.

Proof: Without loss of generality, we restrict $\mathbf{v}^*$ to its active dimensions ($v_j^* \neq 0$), with active stoichiometric matrix $A$. Then this reduced $\mathbf{v}^*$ is the optimal solution for the following non-linear optimization problem over all concentration vectors $\mathbf{y} \equiv [P, \mathbf{a}]^T \in \mathbb{R}_{>0}^{m+1}$:

$$\underset{\mathbf{y}}{\text{maximize}} \quad \mu$$
$$\text{subject to :}$$
$$A\mathbf{v} = \mu\mathbf{y}$$
$$\forall j : v_j = p_j k_j(\mathbf{a}) \tag{24}$$
$$P = \sum_j p_j$$
$$\rho \geq P + \sum_\alpha a_\alpha.$$

Let $\mathbf{y}^* = [P^*, \mathbf{a}^*]^T$ be the concentrations and $\mu^*$ the growth rate of the optimal solution $\mathbf{v}^*$. Now let us consider a linearized version of this optimization problem, where me maximize the production rate $v_{\mathrm{bio}}$ at constant biomass composition $\mathbf{y}^*$ and effective kinetic constants $k_j^{(\mathrm{eff})} \equiv k_j(\mathbf{a}^*)$ (see Definition 2):

$$\underset{\mathbf{v}}{\text{maximize}} \quad v_{\mathrm{bio}}$$
$$\text{subject to :}$$
$$A\mathbf{v} = v_{\mathrm{bio}}\mathbf{y}^*$$
$$\forall j : v_j = p_j k_j^{(\mathrm{eff})} \tag{25}$$
$$P^* \geq \sum_j p_j.$$

We relaxed the constraint (13) on total protein into an inequality constraint, so that Eq. (25) describes a protein-constrained FBA problem for the active stoichiometric matrix. This is precisely the type of constrained flux balance problem analyzed in refs. [28,29], which prove that the solutions $\mathbf{v}^{\mathrm{opt}}$ to the optimization problem defined by Eq. (25) are EFMs.

In the optimal solution to the problem defined by Eq. (25), the protein concentration constraint will be active, that is, $P^* = \sum_j p_j$; if not, the biomass production rate $v_{\mathrm{bio}}$ could be increased by multiplying the vector of protein concentrations $\mathbf{p}$ with a constant $>1$ (as $v_j = p_j k_j^*$ for all $j$). Thus, the optimization problem described by Eq. (25) is the same as that described by Eq. (24), except for a reduction in the dimension of the search space due to the fixed concentrations $\mathbf{y}^*$ (Note that the cellular density constraint (14) is trivially respected in Eq. (25) and can be ignored). Accordingly, the flux distribution $\mathbf{v}^*$ that maximizes the balanced growth rate $\mu$ in Eq. (24) also maximizes the biomass production rate $v_{\mathrm{bio}}$ of the protein-constrained FBA problem in Eq. (25); it is hence a representative of an EFM of the active stoichiometric matrix $A$ with biomass $\mathbf{y}^*$[28,29], and thus $\mathbf{v}^*$ is an EGS according to Definition 3. □

In parallel work, de Groot et al.[30] have shown that optimal solutions to balanced growth problems are elementary growth modes (as defined in ref. [30]), and that the active stoichiometric matrix of elementary growth modes has full rank.

If instead of a single constraint on cellular density, multiple density constraints are imposed simultaneously (e.g., to describe separate constraints on different cellular compartments), then the solutions may in some cases correspond to positive linear combinations of EGSs[30,63], and the treatment below needs to be generalized. Multiple density constraints may play a role in the emergence of overflow metabolism in E. coli[54,64], although overflow metabolism can also arise in balanced growth models with a single density constraint[5].

In a dBGS with maximal growth rate for a given active stoichiometric matrix $A$, the cellular components will utilize the full limit on cellular density $\rho$ to saturate enzymes with their substrates. Thus, the constraint (14) will be active, turning the inequality into an equality. The maximal balanced growth rate $\mu^*$ will thus be a function of the maximal cellular density $\rho$. As a reference value for the marginal net benefits of individual concentrations $x_i$, we now define the marginal benefit of the cellular density $\rho$.

Definition 5 (Marginal benefit of the cellular density): In analogy to the marginal net benefits of cellular components, we define the marginal benefit of the cellular density as the fitness increase facilitated by a small increase in $\rho$,

$$\eta_\rho \equiv \frac{1}{\mu^*}\frac{d\mu^*}{d\rho}.$$

We can now relate $\eta_\rho$ to the total marginal net benefits of all concentrations. To do this, we derive necessary conditions for any optimal BGS at constant cellular density $\rho$, using the method of Lagrange multipliers. The Lagrange multipliers quantify the importance of the density constraint, Eq. (14), and of the constraints for the dependent reactants, Eq. (18), for the maximization of the objective function. The Lagrangian $\mathcal{L}$ is a function of $\mathbf{y} = [P, \mathbf{a}]^T$, and $\rho$.

Theorem 10 (Balance equation): In a dBGS with maximal growth rate, the total marginal net benefit of each independent concentration $x_i$ ($i \in \{P, \beta\}$) equals the

marginal benefit of the cellular density $\rho$ scaled by the density factor $\kappa_i$,

$$\forall i \in \{P, \beta\} : \eta_i = \kappa_i \eta_\rho. \qquad (26)$$

Proof: We use the method of Lagrange multipliers to derive necessary conditions for any optimal dBGS at constant cellular density $\rho$. Our objective function is given by Theorem 7, which expresses the growth rate $\mu$ as an explicit function of the concentrations $\mathbf{y} = [P, \mathbf{a}]^T$. The density constraint (14) will be active at maximal growth rate, i.e., it becomes an equality. The density constraint can then be expressed as a function $g_\rho$ that depends on $\rho$ and on the concentrations,

$$g_\rho(P, \mathbf{a}) \equiv P + \sum_\alpha a_\alpha - \rho = 0.$$

Finally, the constraints on each dependent reactant $\gamma$ also only depend on $\mathbf{y} = [P, \mathbf{a}]^T$, with the entries $D_{\gamma P}$ determining the composition of each $\gamma$ in terms of $P$, and $D_{\gamma \beta}$ determining the composition of $\gamma$ in terms of $b_\beta$,

$$g_\gamma(P, \mathbf{a}) \equiv D_{\gamma P} P + \sum_\beta D_{\gamma \beta} b_\beta - c_\gamma = 0.$$

We now define a Lagrangian as the sum of the objective function $\mu$ and the constraints $\mathbf{g}$ scaled by Lagrange multipliers $\lambda_\rho$, accounting for the density constraint (14), and $\lambda_\gamma$, accounting for the dependence of the dependent reactants $\gamma \in \{\gamma\}$, Eq. (18):

$$\mathcal{L} \equiv \mu + \lambda_\rho g_\rho + \sum_\gamma \lambda_\gamma g_\gamma.$$

The first-order necessary conditions for a constrained local maximum are that all partial derivatives of $\mathcal{L}$ with respect to the variables $P$, $b_\beta$, $c_\gamma$ and to the Lagrange multipliers $\lambda_\rho$, $\lambda_\gamma$ are zero,

$$\forall i \in \{P, \beta\} : 0 = \frac{\partial \mathcal{L}}{\partial x_i},$$
$$\forall \gamma : 0 = \frac{\partial \mathcal{L}}{\partial c_\gamma},$$
$$\forall \gamma : 0 = \frac{\partial \mathcal{L}}{\partial \lambda_\gamma},$$
$$0 = \frac{\partial \mathcal{L}}{\partial \lambda_\rho}.$$

For the partial derivative with respect to an independent concentration $x_i$ ($i \in \{P, \beta\}$), we have

$$\frac{\partial \mathcal{L}}{\partial x_i} = \frac{\partial \mu}{\partial x_i} + \lambda_\rho + \sum_\gamma \lambda_\gamma D_{\gamma i} = 0.$$

With Theorem 8, this results in

$$\mu \eta_i^0 + \lambda_\rho + \sum_\gamma \lambda_\gamma D_{\gamma i} = 0. \qquad (27)$$

For the partial derivative with respect to a dependent reactant $c_\gamma$, we have

$$\frac{\partial \mathcal{L}}{\partial c_\gamma} = \frac{\partial \mu}{\partial c_\gamma} + \lambda_\rho - \lambda_\gamma = 0.$$

With Eq. (19), we obtain

$$\lambda_\gamma = \mu \eta_\gamma^0 + \lambda_\rho.$$

Substituting $\lambda_\gamma$ from the last equation into Eq. (27) gives (for $i \in \{P, \beta\}$)

$$\mu \eta_i^0 + \lambda_\rho + \sum_\gamma \left( \mu \eta_\gamma^c + \lambda_\rho \right) D_{\gamma i} = 0.$$

Rearranging results in

$$
\begin{aligned}
0 &= \mu \eta_i^0 + \mu \sum_\gamma D_{\gamma i} \eta_\gamma^c + \lambda_\rho \left( 1 + \sum_\gamma D_{\gamma i} \right) \\
&= \mu \eta_i + \lambda_\rho \kappa_i \\
&= \mu \eta_i - \mu \eta_\rho \kappa_i,
\end{aligned}
\qquad (28)
$$

where we used $\eta_\rho = -\lambda_\rho/\mu$, which follows directly from the envelope theorem[65]. With $\mu > 0$, we can divide by $\mu$ to obtain the balance equation. $\square$

The optimal state is perfectly balanced: the total marginal net benefit of each independent cellular concentration $x_i$ equals the marginal benefit of the cellular density, scaled by $\kappa_i$ to account for its total utilization of cellular density. If $i$ does not have any dependent reactants ($\forall \gamma: D_{\gamma i} = 0$), then the balance equation simplifies to $\eta_i = \eta_i^0 = \eta_\rho$ (Eq. (10)).

Theorem 10 states that if the dry weight density $\rho$ would be allowed to increase by a small amount, such as $1 \, \mathrm{mg \, l^{-1}}$, then the marginal fitness gain that could be achieved by increasing protein concentration (plus dependent concentrations) by this amount is identical to that achieved by increasing the concentration of any reactant $\beta$ (plus its dependent concentrations) by the same amount.

Instead of using Lagrange multipliers in the proof, one could express the total protein concentration $P = \rho - \sum_\alpha a_\alpha$ (constraint (14)) and the dependent reactant

concentrations $c_\gamma = D_{\gamma P} P + \Sigma_\beta D_{\gamma \beta} b_\beta$ (Eq. (18)) in terms of $\rho$ and of the independent reactant concentrations $\mathbf{b}$. Substituting the resulting expressions into the growth equation (Theorem 7) would result in an objective function that depends only on $\rho$ and $\mathbf{b}$, and that is constrained only by the requirement of positive concentrations. While this would lead to the same balance equations as derived in the Lagrange multiplier framework, this formulation misses important insights that can be derived from the Lagrange multipliers themselves.

**Optimal ribosome proteome fraction.** Here we employ a very simple model for translation[38]. It accounts only for the elongation phase, where one catalyst (the ribosome plus bound mRNA, with concentration $R$) converts one substrate (the ternary complex, with concentration $a_T$) into protein, following irreversible Michaelis–Menten kinetics:

$$k_R \equiv k_R(a_T) = k_{cat} \left( \frac{a_T}{a_T + K_m} \right) \qquad (29)$$

with constant maximal ribosome activity $k_{cat}$ (in units of $[\mathrm{time}]^{-1}$) and Michaelis constant $K_m$ (in units of $[\mathrm{mass}][\mathrm{volume}]^{-1}$).

We assume that the model has no dependent reactants ($A = B$) and that the ternary complex is not used in any other reaction. In this case, the same canceling of production costs as in the model depicted in Supplementary Fig. 1a happens, and the balance of net benefits of ternary complex and total protein, $\eta_T = \eta_P$ (Eq. (10)), simplifies to

$$P u_T^R = 1 - \frac{\mu}{k_R(a_T)}$$

with the kinetic benefit of the ternary complex $T$ for the ribosome $R$, $u_T^R$ (Definition 4). Substituting the partial derivative of irreversible Michaelis–Menten kinetics (Eq. (29)), we obtain

$$\frac{R}{a_T(1 + a_T/K_m)} = 1 - \frac{\mu}{k_R}. \qquad (30)$$

Rearranging Eq. (29), we also see that the kinetics determine the concentration $a_T$ uniquely in terms of $v_R$, $R$, $K_m$, and the ribosome's turnover number $k_{cat}$,

$$a_T = \frac{K_m}{\frac{k_{cat} R}{v_R} - 1}.$$

Substituting this into Eq. (30) gives

$$
\begin{aligned}
R &= \left( 1 - \frac{\mu}{k_R} \right) \left[ \frac{K_m}{\frac{k_{cat} R}{v_R} - 1} \left( 1 + \frac{1}{\frac{k_{cat} R}{v_R} - 1} \right) \right] \\
&= \left( 1 - \frac{\mu}{k_R} \right) K_m \left[ \frac{\frac{k_{cat} R}{v_R}}{\left( \frac{k_{cat} R}{v_R} - 1 \right)^2} \right].
\end{aligned}
\qquad (31)
$$

From the ribosome kinetics and mass conservation of proteins, we have

$$R k_R = v_R = \mu P.$$

Thus, substituting $\mu/k_R = R/P$ and $v_R = \mu P$ in Eq. (31), we obtain

$$\frac{R}{P} = \left( 1 - \frac{R}{P} \right) \frac{K_m}{P} \left[ \frac{\frac{k_{cat} R}{\mu P}}{\left( \frac{k_{cat} R}{\mu P} - 1 \right)^2} \right].$$

This is equivalent to a quadratic equation in $R/P$,

$$\left( \frac{R}{P} \right)^2 + \frac{\mu}{k_{cat}} \left( \frac{K_m}{P} - 2 \right) \left( \frac{R}{P} \right) + \left( \frac{\mu}{k_{cat}} \right)^2 \left( 1 - \frac{k_{cat} K_m}{\mu P} \right) = 0. \qquad (32)$$

Its two solutions are

$$\frac{R}{P} = \frac{\mu}{k_{cat}} \left[ 1 + \frac{K_m}{2P} \left( \pm \sqrt{1 + \frac{4P}{K_m} \left( \frac{k_{cat}}{\mu} - 1 \right)} - 1 \right) \right].$$

To see which of the two solutions is relevant, we rewrite this as

$$k_{cat} R = \mu P \left[ 1 + \frac{K_m}{2P} \left( \pm \sqrt{1 + \frac{4P}{K_m} \left( \frac{k_{cat}}{\mu} - 1 \right)} - 1 \right) \right]. \qquad (33)$$

Because $k_{cat} R > R k_R = v_R = \mu P$, the term in square brackets $[ \cdot ]$ in Eq. (33) must be $> 1$. Only the positive root is compatible with this condition. Thus, the ratio $R/P$ is uniquely determined by

$$\frac{R}{P} = \frac{\mu}{k_{cat}} \left[ 1 + \frac{K_m}{2P} \left( \sqrt{1 + \frac{4P}{K_m} \left( \frac{k_{cat}}{\mu} - 1 \right)} - 1 \right) \right].$$

To relate this expression to experimental data, we need to remember that ribosomes consist of protein and RNA. To estimate the ribosome proteome fraction $\phi_R$, we thus need to scale the previous expression by the fraction $r_P$ of

ribosome which is protein, resulting in the final equation

$$\phi_R(\mu) = \frac{\mu r_P}{k_{cat}} \left[ 1 + \frac{K_m}{2P} \left( \sqrt{1 + \frac{4P}{K_m} \left( \frac{k_{cat}}{\mu} - 1 \right)} - 1 \right) \right]. \tag{34}$$

The same procedure can be used to find an equation for $\phi_R$ that ignores the production costs. Starting from Eq. (31) without the production cost term $\mu/k_R$, we obtain

$$\frac{R}{P} \approx \frac{K_m}{P} \left[ \frac{\frac{k_{cat}R}{\mu P}}{\left( \frac{k_{cat}R}{\mu P} - 1 \right)^2} \right],$$

which results in a quadratic equation similar to Eq. (32),

$$\left( \frac{R}{P} \right)^2 - 2 \frac{\mu}{k_{cat}} \frac{R}{P} + \left( \frac{\mu}{k_{cat}} \right)^2 \left( 1 - \frac{k_{cat}K_m}{\mu P} \right) \approx 0.$$

Solving for $R/P$ gives

$$\frac{R}{P} \approx \frac{\mu}{k_{cat}} \left[ 1 \pm \sqrt{\frac{k_{cat}K_m}{\mu P}} \right]. \tag{35}$$

Again because $Rk_{cat} > \mu P$, the term in square brackets [ · ] in Eq. (35) must be >1, and again only the positive root is compatible with this condition. Thus, the ribosome proteome fraction is uniquely determined in this approximation by

$$\phi_R \approx \frac{\mu r_P}{k_{cat}} \left[ 1 + \sqrt{\frac{k_{cat}K_m}{\mu P}} \right]. \tag{36}$$

We compared the predictions for $\phi_R$ to experimental estimates based on quantitative proteomics[45] and on total RNA to protein ratios[19,42,46,66]. While all estimates are very similar (Fig. 2), given that on the order of 20% of total RNA is tRNA[42] and that this proportion is at least moderately growth rate dependent[67], the exact growth rate dependence of $\phi_R$ may be captured more faithfully by the proteomics data.

To calculate $\phi_R$ from the proteomics measurements, we first calculated the mean over all molar concentrations of ribosomal proteins reported by Schmidt et al.[45]. Molar concentrations of the ribosome were converted to mass concentrations by multiplying with molar masses derived from the amino acid sequences for the protein parts and nucleotide sequences for the RNA parts. For this, we assumed that each ribosome contained one copy of each of its constituents, with the exception of four copies of RplL[68]. We multiplied the ribosome mass concentrations with the mass fraction of ribosomes that is protein ($r_P = 0.358$[45]), and divided the result by the total protein mass concentration $P$ to obtain $\phi_R$. The proteome fraction of actively translating ribosomes was determined based on total ribosome proteome fraction and the fraction of active ribosome at different growth rates. The latter was estimated by fitting a smooth saturation function $s(\mu) = \mu/(\mu + z)$ over the fractions of active ribosomes estimated in ref. [46], with the best-fitting parameter $z = 0.124\,h^{-1}$. Non-linear fitting was performed using the function nls() in gnu R[69].

We set the Michaelis constant of the ribosome to $K'_m = 3 \times 10^{-6}\,mol\,l^{-1}$, based on the diffusion limit for ternary complexes calculated in ref. [38]. We set the ribosome's turnover number to $k_{cat} = 22\,AA\,s^{-1}$, the highest elongation rate observed experimentally in ref. [42]. As we do not distinguish between different ternary complexes and the ribosome only accepts one of the 40 different ternary complex types at any given time, $K'_m$ was multiplied by 40 (see ref. [38]), resulting in an effective Michaelis constant of $K_m = 1.2 \times 10^{-4}\,mol\,l^{-1}$. For consistency of the units with the mass concentration units used throughout our paper, the kinetic parameters had to be converted from molar to mass concentrations. The mean weight (±SD) of amino acids across all conditions assayed in ref. [45] was (132.60 ± 0.09) Da; the ribosome molecular weight is 2,306,967 Da; and the mean weight of ternary complexes is (69,167 ± 1351) g mol$^{-1}$. With these numbers, we obtain $k_{cat} = 22\,AA\,s^{-1} \times (132.60\,Da\,AA^{-1})/(2,306,967\,Da) \times 3600\,s\,h^{-1} = 4.55\,h^{-1}$, and $K_m = 40 \times 3 \times 10^{-6}\,mol\,l^{-1} \times 69,167\,g\,mol^{-1} = 8.30\,g\,l^{-1}$. For the predictions based on Eq. (34), we set the total protein concentration to $P = 127.4\,g\,l^{-1}$ [45].

For yeast, the concentration of actively translating ribosomes was determined based on total ribosome concentration and the fraction of active ribosome at different growth rates; the data was extracted from the figures of ref. [47] using the GetData Graph Digitizer program (Version 2.26, obtained from http://getdata-graph-digitizer.com/). The fraction of active ribosomes was estimated by fitting a smooth saturation function $s(\mu) = \mu/(\mu + z)$ over the fractions of active ribosomes estimated in ref. [47], again using the nls() function in R. The best-fitting parameter was $z = 0.122\,h^{-1}$, very close to the E. coli estimate. We again set $K'_m$ to the diffusion limit[38] $K_m = 3 \times 10^{-6}\,mol\,l^{-1}$, multiplied with the number of different ternary complexes, of which there are 41 in yeast[70]. The ribosome's turnover number was set to $k_{cat} = 10\,AA\,s^{-1}$, the highest elongation rate observed experimentally according to ref. [71]. To convert to mass units, we used the mean weight of amino acids (130 Da)[72], the ribosome molecular weight 3,620,000 Da[73], and the molecular weight of ternary complex (240,000 Da)[74–76]. With these numbers, we obtain $k_{cat} = 10\,AA\,s^{-1} \times (130\,Da\,AA^{-1})/(3,620,000\,Da) \times 3600\,s\,1\,h^{-1} = 1.29\,h^{-1}$, and $K_m = 41 \times 3 \times 10^{-6}\,mol\,l^{-1} \times 240,000\,g\,mol^{-1} = 29.52\,g\,l^{-1}$. In yeast, the mass fraction of ribosomes that is protein is $r_P = 0.45$[73]. For the

predictions based on Eq. (34), we set the total protein concentration to the haploid cell value $P = 85.7\,g\,l^{-1}$ [77].

To quantify the fit of our predictions for $\phi_R$ to the observed ribosomal proteome fractions, we calculated Pearson's correlation coefficient $r$ between observed and predicted values as well as the coefficient of determination

$$R^2 \equiv 1 - \frac{SS_{res}}{SS_{tot}}$$

with the total sum of squares $SS_{tot} = \sum_i (\phi_{R,i} - \bar{\phi}_R)^2$ (proportional to the variance of the data) and the residual sum of squares $SS_{res} = \sum_i (\phi_{R,i} - \phi_{R,i}^{predicted})^2$ (proportional to the variance of the residuals).

**Dependence of maximal growth rate on cellular water content.** Cayley et al.[40,50] have shown that the internal water content of E. coli cells increases when these are grown in environments with reduced osmolarity. This effect corresponds to a decrease of cellular dry weight per volume, $\rho$, by $\delta\rho$. $\eta_\rho$ quantifies the associated reduction in relative fitness, $\delta f = \delta\mu^*/\mu^* = \eta_\rho \delta\rho$, with $\mu^*$ the maximal growth rate (Definition 5). The relative change in the maximal growth rate per relative change in $\rho$ is then

$$\frac{d\ln\mu^*}{d\ln\rho} = \frac{\rho}{\mu^*} \frac{d\mu^*}{d\rho} = \rho\eta_\rho \tag{37}$$

From Eq. (26), we know that $\eta_\rho = \kappa_\rho \eta_\rho$; if there are no dependent reactants for $P$ (i.e., $\forall\gamma: D_{\gamma P} = 0$), this simplifies to

$$\eta_\rho = \eta_P^0 = \frac{1}{P} - \sum_j q_P^j, \tag{38}$$

and thus

$$\frac{\rho}{\mu^*} \frac{d\mu^*}{d\rho} = \rho\eta_\rho = \rho \left( \frac{1}{P} - \sum_j q_P^j \right). \tag{39}$$

The mass fraction of total protein in cell dry weight $P/\rho \approx 0.55$ has been shown to be approximately constant for E. coli across growth conditions supporting intermediate to high growth rates[40,45,49].

To estimate the total protein production cost $\sum_j q_P^j$, we consider the simplest possible whole-cell model, comprising only a transport reaction and the ribosome reaction (Supplementary Fig. 2). The active stoichiometric matrix $A$ of this model and its inverse $A^{-1}$ are (written here with row and column labels):

$$A = \begin{matrix} & t & R \\ 1 & \\ P & \end{matrix} \begin{bmatrix} 1 & -1 \\ 0 & 1 \end{bmatrix}, \qquad A^{-1} = \begin{matrix} & 1 & P \\ t & \\ R & \end{matrix} \begin{bmatrix} 1 & 1 \\ 0 & 1 \end{bmatrix}.$$

The density is determined only by its two components,

$$\rho = P + a_1,$$

where

$$P = p_t + p_R.$$

From the inverse $A^{-1}$ and Theorem 5, we obtain

$$v_t = \mu(P + a_1) = \mu\rho \tag{40}$$

and

$$v_R = \mu P. \tag{41}$$

From the inverse $A^{-1}$ and Eq. (23), we get

$$\sum_j q_P^j = \frac{1}{P} \left( \frac{\mu}{k_t} + \frac{\mu}{k_R} \right) = \frac{1}{P} \left( \frac{\mu p_T}{v_t} + \frac{\mu p_R}{v_R} \right).$$

Combining this with Eqs. (40) and (41) and using $\phi_R = p_R/P$ and $\phi_t = p_t/P = 1 - \phi_R$, we obtain

$$\sum_j q_P^j = \frac{1}{P} \left( \frac{\mu p_t}{\mu\rho} + \frac{\mu p_R}{\mu P} \right)$$
$$= \frac{(1 - \phi_R)}{\rho} + \frac{\phi_R}{P}.$$

Inserting this in Eq. (39) results in

$$\rho\eta_\rho = \rho \left( \frac{1}{P} - \frac{(1 - \phi_R)}{\rho} - \frac{\phi_R}{P} \right)$$
$$= \frac{\rho}{P} - 1 + \phi_R - \frac{\rho}{P} \phi_R \tag{42}$$
$$= \left( \frac{\rho}{P} - 1 \right) (1 - \phi_R).$$

The growth rate in the reference growth condition of osmolarity Osm = 0.28 in ref. [50] is $\mu = 1.0\,h^{-1}$. From Eq. (34), we estimate the mass fraction of ribosomal proteins in total protein $\phi_R$ at this growth rate as $\phi_R = 0.19$. Substituting this value into Eq. (42) together with $P/\rho = 0.55$, we estimate the relative change in the maximal growth rate per relative change in $\rho$ as

$$\rho\eta_\rho = 0.66.$$

Note that instead of a density constraint on total dry mass $\rho$, previous analyses of

schematic and coarse-grained models of balanced growth[3,5–9,19] utilized a constraint only on the concentration of macromolecules $P$. Calculating $P\eta_P$ instead of $\rho\eta_\rho$ leads to a replacement of the factor $(\rho/P - 1)$ with $(1 - P/\rho)$ compared with the last line of Eq. (42), and the same parameterization then leads to a prediction of $P\eta_P = 0.36$.

Cayley et al.[50] report cell growth at reduced osmolarities, summarized in Supplementary Table 1. The cell-free water content $\overline{V}_{free}$ in Supplementary Table 1 is calculated from the total cell water $\overline{V}_{cell}$ minus the observed constant bound water $\overline{V}_b = 0.40 \pm 0.04$ ml gCDW$^{-1}$[40]. Errors are estimated standard deviations based on error propagation among normally distributed random variables. Supplementary Fig. 3 plots the natural logarithms of $\mu$ and $\rho$. Linear regression over the three available data points results in an estimated slope of 0.66, indistinguishable from our estimate of $\frac{d\ln\mu^*}{d\ln\rho} = \rho\eta_\rho = 0.66$.

**Reporting summary**. Further information on research design is available in the Nature Research Reporting Summary linked to this article.

## Data availability
The datasets used for Fig. 2 and Supplementary Figs. 3 and 4 are available from the original sources (refs. [19,42,45–47,50,66]).

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

## Acknowledgements

We thank Johannes Berg, Oliver Ebenhöh, Daan de Groot, Xiao-Pan Hu, Terry Hwa, Michael Lässig, Wolfram Liebermeister, Elad Noor, and Deniz Sezer for discussions. We thank Xiao-Pan Hu for help with the translation model and the calculation of active ribosome fractions, and Jin Wang for help with assembling the yeast parameters. This work was funded by the German Academic Exchange Service (DAAD) through a fellowship (IRTG 1525) to H.D. and by the Deutsche Forschungsgemeinschaft (DFG, German Research Foundation) through grants IRTG 1525, CRC 680, CRC 1310, and, under Germany's Excellence Strategy, through grant EXC 2048/1 (Project ID: 390686111).

## Author contributions

H.D. and M.J.L. jointly conceived the study, interpreted the results, and wrote the manuscript. H.D. developed the GBA framework, performed all data analyses, and derived all formal results except Theorems 1–3, which were derived by MJL.

## Competing interests

The authors declare no competing interests.
