## [Peer Review File · Nature Communications]

Reviewers' comments:

Reviewer #1 (Remarks to the Author):

The paper by Dourado and Lercher proposes a theory of balanced cellular growth that takes a resource allocation perspective on flux balance analysis, like in other recent work, in order to improve predictions of the growth rate under the assumptions that microorganisms have evolved to optimize the growth rate. The work is thoughtful and well-written, making multiple interesting connections to other modeling frameworks like FBA and MCA.

Probably the most original contribution is to show that growth rate optimization requires satisfaction of a necessary condition given by Eq. 10, obtained via the method of Lagrange multipliers.

These positive comments have to be balanced against the following three issues that concern the presentation of the work (point 1) and its practical relevance (points 2 and 3).

1. The paper aims at presenting an all-encompassing theory with many formal definitions and many connections to other theoretical frameworks, which makes the paper hard to read, even for someone familiar with the topic. In particular, the main text reads as an extended abstract of the Supplementary Information and the contents cannot be really understood without reading the latter in detail. On quite a few points, I have had to read the SI carefully to understand what the authors were doing or claiming, for example:

- The notions of BGS and EGS are introduced in two phrases and the reduction of the study of optimal BGSs to the study of EGS is posited without explanation. It also leads to a subtle shift in the meaning of the symbols, as for example the matrix A seems to refer to the active stoichiometry matrix of the entire system in Eq. 1 and to that of an EGS in Eq. 4 (by the way, replacing the inverse of A by I is a bit unfortunate as the latter symbol is often used for the identity matrix).

- I found the definition of marginal costs/benefits difficult to understand from the discussion in the main text, and did not follow how the expressions for η_P and η_α were derived, and how Eq. 8 is obtained.

More generally, the question can be asked whether the format of a short Nat Commun paper is the best way for presenting a complex theoretical framework building upon so many definitions and arguments.

2. There is a discrepancy between the claimed generality of the framework and the applications to illustrate its practical usefulness. In the end, the main application consists of a "whole-cell model" consisting of a linear pathway of two reactions to account for a growth law that has been explained already by other modeling schemes that the authors refer as dealing with "toy models". What is really missing is a convincing demonstration of the usefulness of the framework on a larger model with interesting novel insights provided by the study. Could the models in Refs 13 or 14 be used as a starting-point for this?

3. If this framework is to be an alternative to FBA, as the authors suggest in Figure 1, it needs to confront head-on the difficult problems of building large kinetic models with sensible parameter values and the numerical resolution of large nonlinear optimization problems. The authors remain silent on these points; a casual reference in the Discussion to "artificial intelligence" supplying missing parameter values is not enough in this context.

In conclusion, the paper presents an interesting and promising theoretical framework, but the condensed format of the presentation does not really allow to explain well the specificities of the approach and a convincing application of its added value with respect to other existing approaches is currently missing.

Reviewer #2 (Remarks to the Author):

In this paper the authors present a mathematical framework for modeling cellular growth and metabolism called "Growth Balance Analysis (GBA)", which integrates metabolite, enzyme levels, reaction fluxes, organism growth rates, and a capacity constraint (i.e., an upper bound on total osmolyte density) in the steady-state mass balance characteristic of FBA. Under the GBA framework, metabolite and protein levels are state variables whereas growth rate and reaction fluxes are cast as response variables.

Derivation using null space analysis of stoichiometry matrix and capacity constraint leads to the formulation of explicit relationships between cellular variables. In addition, by calculating the "marginal net benefit" accorded by metabolite and protein perturbations and using Lagrange multipliers in growth optimization, the authors are able to establish that at optimal growth the marginal benefit of every independent metabolite is balanced. Quantitative predictions made by the authors include the variation of active ribosomal protein fraction and cellular density with growth rate in *E. coli*, both of which show good agreement with experimental measurements. The crux of the formulation lies in considering a subset of the stoichiometric matrix consisting of all flux-carrying reactions (dubbed the "active stoichiometric matrix"), which enables the authors to determine the impact of individual reaction fluxes on the growth-dictated dilution of cellular components.

GBA provides useful insight of the cellular state but requires a significant investment in experimental measurables such as metabolite and protein concentrations, knowledge of kinetic laws and parameters. Unfortunately, all these datasets are seldom available for the same organism and condition. Overall, the work of Dourado and Lercher is of sound scientific merit and is a significant contribution to the field of metabolic modeling. The authors are asked to consider the following points:

Major concerns

In the Introduction some background on kinetic modeling should also be discussed, as the issues faced in kinetic modeling such as knowledge of kinetic laws and parameters are also relevant to GBA. How applicable are kinetic parameterization algorithms to GBA modeling? Why kinetic modeling algorithms cannot be deployed to solve the problem instead of GBA?

Without careful examination of and working experience of other modeling frameworks, it might not be clear to the reader what GBA can offer in comparison. Figure 1 does a good job but only for GBA vs. FBA. The authors need to do a more thorough job comparing and contrasting GBA with other metabolic frameworks that rely on some type of a resource constraint such as by Molenaar et al. (considers capacity constraints, similar to GBA), ME-models (consider enzyme synthesis and efficiency), and kinetic model (considers kinetic laws, and substrate and enzyme levels). It is unclear what is the value-proposition of GBA in comparison with existing frameworks.

A key insight of GBA is the insight that the inverse (I) of the active stoichiometric matrix can be used to quantify metabolite dilution by reaction fluxes, however, there was no case study set up to use this inverse I . Thus, we recommend that either GBA be applied and the final solutions be compared to existing metabolic models with available kinetic expressions or a discussion be included so as to illustrate the relevance of incorporating I in metabolic models.

On page 2, please elaborate on the "first principles" in "... demand for catalytic proteins from first principles...?"

Is there a typo on page 7, equation 10 where the term κ_i is missing? Or is there an omission regarding i not having any dependent reactants (i.e., $D_{\gamma i} = 0, \forall \gamma$)?

The authors should mention why data from (Schmidt et al., 2016) (ref 32) and not (Klumpp et al., 2013) (ref 26) was used to construct Figure 2, when the translation model was taken from (Klumpp et al., 2013) and both the papers contain proteomic measurements in *E. coli*.

As GBA is being presented as a generalized modeling framework, it would add to its merit if it can explain proteomic allocations in varied systems, such as ribosomal and metabolic protein fractions in yeast (Paulo et al., 2015, 2016; Metzl-Raz et al., 2017). Additionally, it might be interesting to explore how well GBA does in explaining cases where ribosomal concentrations are not optimally tuned to cellular growth (unlike *E. coli* which is the sole case being explored).

The manuscript will benefit from including a section on “GBA implementation on genome-scale models”. The authors present FBA as a linearized version of GBA, but replacing growth-rate optimization with the search for the solution set of GBA-derived balance equations is complicated by 1) lack of rate expressions and in vivo kinetic parameters for a majority of biochemical reactions, and 2) a priori knowledge of the subset of flux-carrying metabolic reactions (which will also depend on growth conditions) to construct the “active stoichiometric matrix”.

Minor concerns

Page 2, paragraph 2: recommend changing “all cellular components must be reproduced in proportion...” to “all cellular components must be produced in proportion...”.

In the Abstract, “experimentally testable predictions” are mentioned but never specified further.

References

Klumpp, S. et al. (2013) ‘Molecular crowding limits translation and cell growth’, *Proceedings of the National Academy of Sciences of the United States of America*. doi: 10.1073/pnas.1310377110.

Metzl-Raz, E. et al. (2017) ‘Principles of cellular resource allocation revealed by condition-dependent proteome profiling’, *eLife*. doi: 10.7554/eLife.28034.

Paulo, J. A. et al. (2015) ‘Proteome-wide quantitative multiplexed profiling of protein expression: Carbon-source dependency in *Saccharomyces cerevisiae*’, *Molecular Biology of the Cell*. doi: 10.1091/mbc.E15-07-0499.

Paulo, J. A. et al. (2016) ‘Quantitative mass spectrometry-based multiplexing compares the abundance of 5000 *S. cerevisiae* proteins across 10 carbon sources’, *Journal of Proteomics*. doi: 10.1016/j.jprot.2016.07.005.

Schmidt, A. et al. (2016) ‘The quantitative and condition-dependent *Escherichia coli* proteome’, *Nature Biotechnology*. doi: 10.1038/nbt.3418.

Reviewer #1

The paper by Dourado and Lercher proposes a theory of balanced cellular growth that takes a resource allocation perspective on flux balance analysis, like in other recent work, in order to improve predictions of the growth rate under the assumptions that microorganisms have evolved to optimize the growth rate. The work is thoughtful and well-written, making multiple interesting connections to other modeling frameworks like FBA and MCA.

Probably the most original contribution is to show that growth rate optimization requires satisfaction of a necessary condition given by Eq. 10, obtained via the method of Lagrange multipliers.

Response: We thank the Reviewer for this positive evaluation.

Specific comments

These positive comments have to be balanced against the following three issues that concern the presentation of the work (point 1) and its practical relevance (points 2 and 3).

- 1. The paper aims at presenting an all-encompassing theory with many formal definitions and many connections to other theoretical frameworks, which makes the paper hard to read, even for someone familiar with the topic. In particular, the main text reads as an extended abstract of the Supplementary Information and the contents cannot be really understood without reading the latter in detail. On quite a few points, I have had to read the SI carefully to understand what the authors were doing or claiming, for example:
 - The notions of BGS and EGS are introduced in two phrases and the reduction of the study of optimal BGSs to the study of EGS is posited without explanation. It also leads to a subtle shift in the meaning of the symbols, as for example the matrix A seems to refer to the active stoichiometry matrix of the entire system in Eq. 1 and to that of an EGS in Eq. 4 (by the way, replacing the inverse of A by I is a bit unfortunate as the latter symbol is often used for the identity matrix).*
 - I found the definition of marginal costs/benefits difficult to understand from the discussion in the main text, and did not follow how the expressions for η_P and η_α were derived, and how Eq. 8 is obtained.*More generally, the question can be asked whether the format of a short Nat Commun paper is the best way for presenting a complex theoretical framework building upon so many definitions and arguments.*

Response: We agree that the main text only provides a summary of the main ideas and results; for anyone who wants to deeply understand the formalism, a careful reading of *SI text A* is crucial. The main text is written predominantly to provide an overview of the rationale of GBA and of its potential applications, aimed at a general audience that may be neither trained in nor deeply interested in the mathematical details. We believe that this type of “division of labor” between main text and SI is characteristic for many publications in the *Nature* family journals, and should not be seen as a negative. We do, however, fully agree that some parts of the main text can and should be made more comprehensible for the benefit of all readers.

Action: We rewrote and expanded large parts of the main text, improving the presentation to make it more easily understandable without having to resort to the SI. We paid particular attention to the points listed by the Reviewer (Def. of BGS and EGS, line 125ff.; relationship between optimal states and EGS, line 132ff.; restriction to full-rank active matrices A , line 134ff.). We also replaced the symbol for the inverse of the active stoichiometric matrix, which was I , with the standard symbol, A^{-1} , throughout the manuscript and SI.

We extensively rewrote the description of marginal costs and benefits, both in the main text (the subsection beginning on line 166) and in the SI (the subsection beginning on line 768). The derivation of Eq. (8) is now explained in detail in the paragraph beginning on line 792, and the following paragraphs detail the derivation of η_P and η_α .

We still keep the full formal treatment to the SI, but now strive to give enough detail in the main text to provide a general idea of the necessary steps for each derivation. This not only makes the main text easier to understand for a general audience, but also provides an overview to the expert reader before he or she dives more deeply into the mathematical details.

2. *There is a discrepancy between the claimed generality of the framework and the applications to illustrate its practical usefulness. In the end, the main application consists of a "whole-cell model" consisting of a linear pathway of two reactions to account for a growth law that has been explained already by other modeling schemes that the authors refer as dealing with "toy models". What is really missing is a convincing demonstration of the usefulness of the framework on a larger model with interesting novel insights provided by the study. Could the models in Refs 13 or 14 be used as a starting-point for this?*

Response: The manuscript describes two applications. The first application examines the growth rate dependence of ribosome concentrations (Fig. 2). The derivation of this exact relationship does not depend on a specific (linear or non-linear, small or large) model, but is valid for *any* whole-cell balanced growth model in which the ribosome consumes a single substrate type to make proteins. Thus, our derivation is much more general than any previous modeling scheme addressing this issue, all of which were indeed linear models with 2-7 reactions (Refs. 3, 5-9). While these small models were able to predict an approximately linear relationship between ribosomal protein fraction and growth rate, their description of experimental observations was qualitative only (or had to rely on fitting multiple parameters to the data to achieve a quantitative description). In contrast, our general model makes accurate quantitative predictions for *E. coli* based only on known ribosome kinetics. In the revised version, we additionally perform the same calculation for yeast, again with a very good fit to experimental data without fitting any parameters (Fig. 2b). Thus, our first and major application does not rely on a specific (small, linear) model, and it makes quantitative predictions for *E. coli* and yeast not possible using previously published frameworks.

The second application concerns the effects of reduced dry mass density on optimal growth (Fig. S3). Here, we indeed use a linear ("toy") model with only two reactions. While this prediction would thus have been possible from an analytical optimization without using our general framework, only the inclusion of metabolite concentrations in the capacity constraint (which was not deemed necessary in the previous treatments of Refs. 3, 5-9, 19) results in the predicted dependence. Our predictions agree quantitatively with the limited available experimental data in Fig. S3; an otherwise identical model that limits only the total protein concentration results in a much lower estimate for the dependence of growth rate on the cellular capacity. Our results not only emphasize the biological significance of the capacity

constraint, but – in contrast to models that only limit total protein – provide an explanation for the observation that *E. coli*'s dry mass density is roughly constant across conditions (Refs. 43-45).

In sum, although we did not make this sufficiently clear in the previous manuscript, we indeed demonstrated the application to an arbitrarily large model (using a prediction for which all terms containing unknown kinetic parameters cancel from the final equation), and we explained two experimental observations that were previously not understood in quantitative terms. In addition, our mathematical derivations provide general insights into cellular resource allocation, such as the relationship between marginal fitness costs and protein expression, Eq. (8).

We fully agree with the referee that the application of our framework to a genome-scale model of bacterial growth is highly desirable, and our group is actively working on such an application. However, reaching this goal will require years of work. While existing constraint-based genome-scale models, such as Bernard Palsson's ME models or Anne Goelzer's and Vincent Fromion's RBA models, could provide a basis for this endeavor, these do not include the dependence of reaction fluxes on metabolite concentrations. Thus, as pointed out by the Reviewer in Comment 3, the most important obstacles that need to be overcome on the way to genome-scale GBA applications are the full, non-linear kinetic parameterization and the numerical solution of the resulting optimization problem (with the aid of the balance equations). These goals are outside the scope of the current manuscript; here, we develop the mathematical foundations and demonstrate their usefulness for specific predictions even in the absence of a more detailed explicit model. We regret that we did not make the distinction between the goals of our manuscript and the challenges still lying ahead sufficiently clear in the previous version.

Action: We extended the discussion of the two applications to clarify the underlying assumptions and to emphasize the differences to previous work, as detailed in the Response (line 235ff., line 245ff.); we also no longer refer to small, schematic models as “toy” models. In addition, we now clearly state the goals of the present study (line 28ff., line 334ff.) – the derivation of a rigorous analytical framework for GBA, which provides a mathematical basis for current applications to smaller (“toy”) models as well as for future coarse-grained and genome-scale applications. Finally, we lay out the challenges that need to be overcome before GBA may eventually replace FBA at least for model organisms (line 291ff.; see also our response/action to Comment 3).

- 3. If this framework is to be an alternative to FBA, as the authors suggest in Figure 1, it needs to confront head-on the difficult problems of building large kinetic models with sensible parameter values and the numerical resolution of large nonlinear optimization problems. The authors remain silent on these points; a casual reference in the Discussion to "artificial intelligence" supplying missing parameter values is not enough in this context.*

Response: As demonstrated by our applications (see also our response to Comment 2), the GBA scheme already provides important insights and makes testable predictions even without access to a detailed genome-scale kinetic parameterization. Moreover, GBA is a powerful tool for the analysis of schematic (or coarse-grained) models, and would have greatly benefited all previous theoretical studies of balanced growth (e.g., Refs. 3, 5-10). We foresee that GBA will indeed become an exciting alternative to FBA in the future at least for model organisms and synthetic biology “chassis” strains. However, as pointed out by the Reviewer and mentioned in our response to Comment 2, the genome-scale kinetic parameterization and the

non-linear optimization are difficult problems that first need to be tackled, but that lie outside of the scope of the current manuscript. We regret that in the previous version, we did not make our aims more explicit and did not discuss sufficiently what challenges still lie ahead.

Action: In the revised manuscript, we clearly state the goals of the present study (line 28ff., line 334ff.): to provide a rigorous mathematical basis for the analysis of self-replicator models and to demonstrate its general utility. We added a detailed discussion of the application of GBA to genome-scale models (the two paragraphs following line 291), where we clearly list the remaining obstacles that need to be overcome before GBA can replace FBA for realistic, genome-scale models of cellular growth, and we discuss in detail how these obstacles might be overcome.

4. *In conclusion, the paper presents an interesting and promising theoretical framework, but the condensed format of the presentation does not really allow to explain well the specificities of the approach and a convincing application of its added value with respect to other existing approaches is currently missing.*

Response: As outlined in our response to Comment 1, we have expanded and improved the main text to provide a better (and more understandable) overview over the mathematical derivations presented in detail in the SI (which we also improved substantially, in particular in section A.3). As explained in our response to Comment 2, the two applications lead to quantitative predictions not possible with previous methodologies, clearly demonstrating the added value of GBA over alternative approaches (even if this was not made sufficiently clear in the previous manuscript). We believe that our actions in response to Comments 1-3 have substantially improved the manuscript, and have rectified the problems pointed out by the Reviewer.

Reviewer #2

In this paper the authors present a mathematical framework for modeling cellular growth and metabolism called “Growth Balance Analysis (GBA)”, which integrates metabolite, enzyme levels, reaction fluxes, organism growth rates, and a capacity constraint (i.e., an upper bound on total osmolyte density) in the steady-state mass balance characteristic of FBA. Under the GBA framework, metabolite and protein levels are state variables whereas growth rate and reaction fluxes are cast as response variables.

*Derivation using null space analysis of stoichiometry matrix and capacity constraint leads to the formulation of explicit relationships between cellular variables. In addition, by calculating the “marginal net benefit” accorded by metabolite and protein perturbations and using Lagrange multipliers in growth optimization, the authors are able to establish that at optimal growth the marginal benefit of every independent metabolite is balanced. Quantitative predictions made by the authors include the variation of active ribosomal protein fraction and cellular density with growth rate in *E. coli*, both of which show good agreement with experimental measurements. The crux of the formulation lies in considering a subset of the stoichiometric matrix consisting of all flux-carrying reactions (dubbed the “active stoichiometric matrix”), which enables the authors to determine the impact of individual reaction fluxes on the growth-dictated dilution of cellular components.*

GBA provides useful insight of the cellular state but requires a significant investment in experimental measurables such as metabolite and protein concentrations, knowledge of kinetic laws and parameters. Unfortunately, all these datasets are seldom available for the same organism and condition. Overall, the work of Dourado and Lercher is of sound scientific merit and is a significant contribution to the field of metabolic modeling.

Response: We thank the Reviewer for this positive evaluation.

Major concerns

The authors are asked to consider the following points:

- 1. In the Introduction some background on kinetic modeling should also be discussed, as the issues faced in kinetic modeling such as knowledge of kinetic laws and parameters are also relevant to GBA. How applicable are kinetic parameterization algorithms to GBA modeling? Why kinetic modeling algorithms cannot be deployed to solve the problem instead of GBA?*

Response: We regret that we omitted a discussion of kinetic modeling approaches in the previous version. Briefly, GBA is based on a global optimization of cellular resource allocation, and thus predicts all concentrations and fluxes based solely on the environmental and physico-chemical boundary conditions. In contrast, kinetic modeling approaches require knowledge of enzyme concentrations, and thus cannot predict resource allocation from first principles. Moreover, as GBA is based on a global optimization of cellular resource allocation, it will be less sensitive than kinetic modeling algorithms to regulatory interactions and to small parameter changes.

Fitting-based parameterization algorithms as those developed for kinetic modeling approaches could indeed be applied to GBA; however, this would to some extent go against our aim to achieve a mechanistic understanding of cellular resource allocation and physiology. Experiences from kinetic modeling, such as the robustness of predictions to alternative parameterizations and the utility of parameter balancing approaches, will benefit the development of fully parameterized GBA models.

Action: As suggested by the Reviewer, we have added a discussion of kinetic modeling methods to the Introduction (line 64ff.), and we discuss the potential utility of existing parameterization algorithms (line 315ff.). We also added a discussion of the major differences between GBA and kinetic model applications (line 325ff.).

2. *Without careful examination of and working experience of other modeling frameworks, it might not be clear to the reader what GBA can offer in comparison. Figure 1 does a good job but only for GBA vs. FBA. The authors need to do a more thorough job comparing and contrasting GBA with other metabolic frameworks that rely on some type of a resource constraint such as by Molenaar et al. (considers capacity constraints, similar to GBA), ME-models (consider enzyme synthesis and efficiency), and kinetic model (considers kinetic laws, and substrate and enzyme levels). It is unclear what is the value-proposition of GBA in comparison with existing frameworks.*

Response: A more thorough comparison of GBA to alternative modeling schemes beyond FBA is indeed desirable.

Action: As suggested by the Reviewer, we added a detailed comparison of GBA to other modeling frameworks, including not only FBA, but also the self-replicator model by Molenaar et al. (Ref. 5), line 275ff.; RBA (Ref. 14) and ME-models (Ref. 15), line 42ff.; and kinetic models (Refs. 24, 25), line 325ff..

3. *A key insight of GBA is the insight that the inverse (I) of the active stoichiometric matrix can be used to quantify metabolite dilution by reaction fluxes, however, there was no case study set up to use this inverse I . Thus, we recommend that either GBA be applied and the final solutions be compared to existing metabolic models with available kinetic expressions or a discussion be included so as to illustrate the relevance of incorporating I in metabolic models.*

Response: The mass balance of biochemical reactions is usually described mathematically through the stoichiometric matrix A , as in $Av=dx/dt$. If A has full column rank (which is the case for elementary flux modes of linear models, Ref. 32, used in Theorem 4), then we can use $I\equiv A^{-1}$ to instead describe the fluxes as a function of the concentration changes that they cause (note that upon suggestion of Reviewer 1, we have replaced I by A^{-1} throughout the manuscript and SI). This insight allows us to express the growth rate as well as the individual protein concentrations as functions of the state variables (reactant and total protein concentrations).

More explicitly, as seen from Eq. (4), A^{-1}_{ji} quantifies what proportion of reaction flux j contributes to the dilution of the downstream cellular component i , providing important information on cellular efficiency. The expression using A^{-1} to quantify the growth rate as a function of cellular concentrations provides important insights into how cellular concentrations affect growth (and hence fitness). Moreover, A^{-1} forms the basis for the calculation of marginal costs and benefits, accounting for all systems-level contributions. In our manuscript, we use the balance of the marginal net benefits at optimal growth to derive a

quantitative estimate of the growth-rate dependence of ribosome concentrations based only on known ribosome kinetics. While our manuscript thus incorporates a number of arguments and examples for the relevance of using A^{-1} in metabolic models, we did not make the centrality of A^{-1} for these sufficiently clear in the text.

Action: We have emphasized the role of A^{-1} throughout the text. In addition, we added a new paragraph to the discussion about the interpretation of A^{-1} , explaining its utility for solving growth models and for understanding cellular efficiency and marginal costs and benefits of cellular resource allocation (line 261ff.).

4. *On page 2, please elaborate on the “first principles” in “... demand for catalytic proteins from first principles...”?*

Response: We apologize for being unclear here. What we meant to say: to support a given metabolic flux, the catalyzing enzyme needs to be present at a concentration that is set by the flux value and the concentrations of the metabolites that enter the reaction’s kinetic function. FBA does not directly account for the need to produce these proteins. Instead, FBA ensures the production of a constant biomass that includes the amino acids needed for growth, which is assumed to be independent of growth conditions and the activity of specific enzymes.

Action: We have clarified this passage (line 35ff.), which now reads: “The resulting computational efficiency comes at the price of ignoring reaction kinetics and the requirement of sufficient enzyme concentrations to catalyze the predicted metabolic fluxes.”

5. *Is there a typo on page 7, equation 10 where the term κ_i is missing? Or is there an omission regarding i not having any dependent reactants (i.e., $D_{\gamma i}=0, \forall \gamma$)?*

Response: On page 5 of the previous version (just above Eq. (4)), we stated that “For clarity of presentation, we here present only the case without dependent reactants; the generalization can be treated similarly and is detailed in SI text A.”. This simplifying assumption was made for the remainder of the main text, and thus we indeed assume $D_{\gamma i}=0, \forall \gamma$, as suspected by the Reviewer. We believe that it is easier to develop an intuition for the main results if we first treat the simpler case without dependent reactants; we thus leave the general expressions, which involve additional terms and symbols, to the SI. We regret that we did not make it sufficiently clear that we kept this assumption for the remainder of the text.

Action: We now emphasize at multiple positions in the manuscript that the main text makes the simplifying assumption that there are no dependent reactants, while the general case is treated in SI text A (line 142ff., line 159ff., line 168ff., line 191ff., line 213ff.).

6. *The authors should mention why data from (Schmidt et al., 2016) (ref 32) and not (Klumpp et al., 2013) (ref 26) was used to construct Figure 2, when the translation model was taken from (Klumpp et al., 2013) and both the papers contain proteomic measurements in E. coli.*

Response: While the paper by Klumpp et al. (now Ref. 39) shows data for the ribosomal protein fraction, this data is not based on proteomics measurements. Instead, the numbers given by Klumpp et al. are derived from the ratio of the dry mass fractions occupied by total RNA and total protein. However, total RNA contains a substantial fraction of non-ribosomal RNA, in particular of tRNA (Ref. 43), and tRNA concentrations increase with growth rate (Ref. 69). In contrast, Schmidt et al. (Ref. 46) directly measured the protein fraction of multiple ribosomal proteins, and thus provide a more reliable quantitative picture of the

growth rate-dependence of the ribosomal protein fraction. This was our rationale for originally only showing the Schmidt *et al.* data. Nevertheless, we agree that it is appropriate to show all available data in the comparison to our predictions.

Action: We have added data from four studies that estimated ϕ_R from the total RNA/protein ratio, including all data shown by Klumpp *et al.* (Fig. 2a and line 967ff.).

7. (a) As GBA is being presented as a generalized modeling framework, it would add to its merit if it can explain proteomic allocations in varied systems, such as ribosomal and metabolic protein fractions in yeast (Paulo *et al.*, 2015, 2016; Metzl-Raz *et al.*, 2017). (b) Additionally, it might be interesting to explore how well GBA does in explaining cases where ribosomal concentrations are not optimally tuned to cellular growth (unlike *E. coli* which is the sole case being explored).

Response: (a) We thank the reviewer for pointing us to proteomic data in yeast that can be used to test our model's predictions also in a eukaryotic model organism. Analogous to the calculation of the optimal ribosomal protein fraction in *E. coli* (Fig. 2a), we now also calculate the optimal ribosomal protein fraction in the yeast *S. cerevisiae* (Fig. 2b). Without fitting any parameters, we again achieve quantitative agreement with the experimental data.

(b) The GBA framework developed in this manuscript assumes growth rate optimization; hence, it cannot successfully predict ribosomal concentrations for organisms in which these are not optimized for growth. To make such predictions, we would need to know the objective function optimized by natural selection (if one indeed exists), which is generally unknown, and extend the GBA framework accordingly.

Action: (a) We have added a panel for yeast to Fig. 2, which compares our prediction of the ribosomal protein fraction with proteomics data acquired across different growth conditions.

(b) We now emphasize that GBA assumes optimal resource allocation, and that it has to be applied with caution to systems that are not optimally tuned for growth; the same is true for environmental or genetic perturbations to which the studied organism has not been adapted by natural selection (line 329ff.).

8. *The manuscript will benefit from including a section on “GBA implementation on genome-scale models”. The authors present FBA as a linearized version of GBA, but replacing growth-rate optimization with the search for the solution set of GBA-derived balance equations is complicated by 1) lack of rate expressions and in vivo kinetic parameters for a majority of biochemical reactions, and 2) a priori knowledge of the subset of flux-carrying metabolic reactions (which will also depend on growth conditions) to construct the “active stoichiometric matrix”.*

Response: We very briefly discussed this in the paragraph starting at the bottom of p.9 of the original submission, but we agree that the importance of this issue warrants a much more detailed discussion (see also our reply to comment 3 by Reviewer #1).

Action: As suggested by the Reviewer, we added a detailed discussion of genome-scale GBA implementations, where we consider potential strategies to parameterize such models and to identify condition-dependent sets of active reactions (the two paragraphs following line 289).

Minor concerns

9. Page 2, paragraph 2: recommend changing “all cellular components must be reproduced in proportion...” to “all cellular components must be produced in proportion...”.

Action: done (line 18).

10. In the Abstract, “experimentally testable predictions” are mentioned but never specified further.

Response: We referred to the two predictions derived in the manuscript and compared to experimental data, the scaling of ribosome concentrations with growth rate (Fig. 2) and the dependence of growth rate on dry weight density (Fig. S3).

Action: We now specify these predictions in the last sentence of the abstract.

REVIEWERS' COMMENTS:

Reviewer #1 (Remarks to the Author):

The authors have changed the text of the manuscript to address the three major comments I raised in the previous round of reviewing.

I continue to think that the real added value of the work is the careful derivation of the framework, which remains mostly hidden in the SI. I am well aware that most Nature papers do not contain much technical material in the main text, but in this case the technical material *is* the main contribution. In the end, however, the authors are right that this is an editorial issue.

The authors have chosen not to develop a more challenging application than a two-step pathway for explaining the quasi-linear relation between growth rate and ribosome mass fraction. Instead they argue that the prediction is quantitative and obtained from the ribosome kinetics in a principled manner without parameter fitting. I can accept this argument although I continue to believe that more challenging examples would have demonstrated the practical usefulness of the framework more convincingly than is done now.

Minor comments

- p11: $\mu < 1$. What are the units? What does this limit correspond to?
- p11: "potentially explaining why... across conditions." I can't follow the reasoning here.
- p9, after Eq. 8: I think that one or two phrases developing the point that "this result provides a formal justification..." would be useful for the reader. The equation tells us, if I understand well, that for constant μ and positive η_i , an increase of x_i allows a decrease of (some) ϕ_j ?
- On the same point, it seems that η_i in Eq. 8 corresponds to η_1^0 in Section A.3, that is, the total marginal benefit reported in the main text used the direct marginal benefit in the appendix?

Reviewer #2 (Remarks to the Author):

The authors did a good job addressing many of the questions raised in our original review. It would have been preferable if a more detailed comparison with resource-allocation based methods such as ME or RBA was made. Also, the kinetic parameterization part is standard but admittedly it does not form the core of the methodology. In light of not delaying things further I am in favor of recommending publication so the community has a chance to apply and test the proposed concepts.

Reviewer #1

The authors have changed the text of the manuscript to address the three major comments I raised in the previous round of reviewing.

- 1. I continue to think that the real added value of the work is the careful derivation of the framework, which remains mostly hidden in the SI. I am well aware that most Nature papers do not contain much technical material in the main text, but in this case the technical material *is* the main contribution. In the end, however, the authors are right that this is an editorial issue.*

Response: We agree that the major contribution made by our work lies in the mathematical derivations. However, their details are too involved for the majority of readers, and thus the Results section only provides a summary. The mathematical details form the methodological part of our paper, and hence they belong into a separate Methods section.

Action: We have moved all derivations from the SI to the new Methods section, so that they now form an integral part of the manuscript.

- 2. The authors have chosen not to develop a more challenging application than a two-step pathway for explaining the quasi-linear relation between growth rate and ribosome mass fraction. Instead they argue that the prediction is quantitative and obtained from the ribosome kinetics in a principled manner without parameter fitting. I can accept this argument although I continue to believe that more challenging examples would have demonstrated the practical usefulness of the framework more convincingly than is done now.*

Response: Our results on the scaling of ribosome concentrations with growth rate are fully general and do not depend on the choice of a specific two-step (or many-step) pathway. More challenging examples – in particular the application to genome-scale bacterial models – will be the focus of future work.

Action: None.

Minor comments

- 3. - p11: $\mu < 1$. What are the units? What does this limit correspond to?*

Response: We apologize for omitting the units – this was supposed to read $\mu < 1 \text{ h}^{-1}$. It is not a limit, it just makes clear which data points we refer to when we talk about “growth on minimal media”.

Action: We added the units.

- 4. - p11: "potentially explaining why... across conditions." I can't follow the reasoning here*

Response: When we ignore the dilution of intermediates, the equation reduces to the simpler equation introduced in Dourado 2017 that predicts the optimal balance of metabolite and enzyme concentrations based solely on the combined mass concentration of metabolites and enzymes. We admit that this was not obvious from the previous, very brief formulation.

Action: We expanded this passage as follows: “In contrast, these approximate predictions are close to observed values for growth on minimal media ($\mu < 1 \text{ h}^{-1}$), indicating that

the dilution of intermediates, μ_{α} , becomes less important at lower growth rates. The latter observation may explain why the relationship between the concentrations of a substrate and its catalysts is well approximated in this regime by simply minimizing their combined mass concentration while keeping the reaction rate constant (cite{Dourado2017}), as this is mathematically equivalent to ignoring the dilution of intermediates.

5. - p9, after Eq. 8: I think that one or two phrases developing the point that "this result provides a formal justification..." would be useful for the reader. The equation tells us, if I understand well, that for constant μ and positive η_i , an increase of x_i allows a decrease of (some) ϕ_j ?

Response: This interpretation is exactly right, and we agree that spelling it out will make the point we want to make much more accessible to the reader.

Action: We added the following explanation immediately after Eq. (8) and before the phrase quoted by the Reviewer: "Thus, for a positive η_i and keeping the growth rate μ constant, a small increase in x_i by Δx_i results in a corresponding reduction of the total protein fraction, $\sum_j \Delta \phi_j = -\eta_i \Delta x_i$: at least some proteins are now required at lower concentrations.

6. - On the same point, it seems that η_i in Eq. 8 corresponds to η_{i^0} in Section A.3, that is, the total marginal benefit reported in the main text used the direct marginal benefit in the appendix?

Response: The Methods section develops the theory for the general case with dependent reactants. In contrast, to facilitate a more intuitive understanding of the theory, the main text provides results only for the simpler case without dependent reactants (as emphasized at multiple places throughout the text). In this simpler case, the direct and total marginal benefits are indeed equal, $\eta_i = \eta_{i^0}$.

Action: To avoid confusion in the careful reader, we now emphasize this fact just before Eq. (8): "(see Methods, section 1.3; note that because here we assume that there are no dependent reactants, direct and total net benefits as defined in Methods are identical)".

Reviewer #2

The authors did a good job addressing many of the questions raised in our original review. It would have been preferable if a more detailed comparison with resource-allocation based methods such as ME or RBA was made. Also, the kinetic parameterization part is standard but admittedly it does not form the core of the methodology. In light of not delaying things further I am in favor of recommending publication so the community has a chance to apply and test the proposed concepts.

Response: We thank the Reviewer for this constructive comment. We believe that a more detailed comparison to ME and RBA is best left to a future paper that examines a genome-scale implementation of GBA.

Action: None.